# Stimulus-selective crosstalk via the NF-κB signaling system reinforces innate immune response to alleviate gut infection

**Balaji Banoth, Budhaditya Chatterjee, Bharath Vijayaragavan, MVR Prasad, Payel Roy, Soumen Basak***

Systems Immunology Laboratory, National Institute of Immunology, New Delhi, India

**Abstract** Tissue microenvironment functions as an important determinant of the inflammatory response elicited by the resident cells. Yet, the underlying molecular mechanisms remain obscure. Our systems-level analyses identified a duration code that instructs stimulus specific crosstalk between TLR4-activated canonical NF-κB pathway and lymphotoxin-β receptor (LTβR) induced non-canonical NF-κB signaling. Indeed, LTβR costimulation synergistically enhanced the late RelA/NF-κB response to TLR4 prolonging NF-κB target gene-expressions. Concomitant LTβR signal targeted TLR4-induced newly synthesized p100, encoded by *Nfkb2*, for processing into p52 that not only neutralized p100 mediated inhibitions, but potently generated RelA:p52/NF-κB activity in a positive feedback loop. Finally, *Nfkb2* connected lymphotoxin signal within the intestinal niche in reinforcing epithelial innate inflammatory RelA/NF-κB response to *Citrobacter rodentium* infection, while *Nfkb2*⁻/⁻ mice succumbed to gut infections owing to stromal defects. In sum, our results suggest that signal integration via the pleiotropic NF-κB system enables tissue microenvironment derived cues in calibrating physiological responses.

**\*For correspondence:** sobasak@nii.ac.in

## Introduction

Tight regulation of inflammatory responses is important; uncontrolled inflammation underlies various human ailments, while insufficient responses limit host defense to pathogens. Tissue-resident cells those that participate in inflammatory immune activation also exhibit functional differences by adapting to the repertoire of cell-differentiating cues present in distinct microenvironments. Indeed, macrophages and dendritic cells present in different anatomic niche display heterogeneity in inflammatory signatures (*Iwasaki and Kelsall, 1999*; *Stout and Suttles, 2004*). Likewise, a requirement for CD40, primarily involved in B-cell maturation, in inflammatory gene expressions in endothelial cells was documented (*Pluvinet et al., 2008*). Similarly, lymph node inducing lymphotoxin-β receptor (LTβR) was shown to be critical for innate immune responses (*Spahn et al., 2004*; *Wang et al., 2010*). Yet, the cellular circuitry that functions at the intersection of tissue microenvironment derived signals and those impinged upon by pro-inflammatory cytokines or pathogen-derived substances remains obscure.

The NF-κB family of transcription factors plays an essential role in activating pathogen-responsive gene-expression program in tissue-resident cells. In the canonical NF-κB pathway, inflammatory cues engage NEMO-IKK2 (NEMO-IKKβ) kinase complex to phosphorylate inhibitory IκB proteins, the major isoform being IκBα, bound to the cytoplasmic RelA:p50 NF-κB dimers. Signal-induced phosphorylation leads to proteasomal degradation of IκBs and release of RelA:p50 dimers into the nucleus. The nuclear RelA dimers activate the expressions of pro-inflammatory chemokine and cytokine genes as well as its own inhibitor IκBα, which ensures proper attenuation of inflammatory responses in a negative feedback loop. In contrast to canonical signaling, the

**eLife digest** The innate immune system is the body's first line of defense against infection and disease. Innate immune cells are found in every tissue type, poised to respond immediately to damaged, stressed, or infected host cells. When innate immune cells recognize any injury or infection, one of the first things they do is trigger the inflammatory response. Fluid and other immune cells then move from the blood into the injured tissues. This movement can cause redness and swelling. But the response helps to establish a physical barrier against the spread of infection, promotes the elimination of both invading microbes and damaged host cells, and encourages the repair of the tissue.

Inflammation is tightly controlled. If the response is too weak, it could leave an individual prone to serious infection. On the other hand, excessive inflammation can severely damage healthy cells and tissues. Inflammation is regulated differently in different tissue types, and the environment within the tissue itself influences the activity of local innate immune cells and the inflammatory response. However, the molecular mechanisms responsible for receiving and interpreting the signals derived from the host tissue remain unknown.

Now, Banoth et al., have revealed that the integration of inflammation-provoking signals, such as injury or infection and cues from the tissue environment occurs via the so-called 'NF-κB signaling system'. NF-κB is a protein found in almost all cell types, and when activated it is able to switch on the expression of many different genes. Banoth et al. explain that signal integration via the NF-κB system enables cues from the tissue environment to tune a cell's responses. Further experiments confirmed the importance of this signal integration by showing how a signal coming from intestinal tissue can influence the activity of innate immune cells and inflammation in the gut.

These findings suggest that a breakdown in the NF-κB signaling system's ability to integrate multiple signals, including those derived from the tissue environment, may be responsible for many inflammatory disorders, and in particular those that involve the gut. Future work is now needed to explore this possibility.

non-canonical pathway transduces signals from cell-differentiating cues those engage BAFFR, CD40, or LTβR. Non-canonical signaling involves NIK and IKK1 (NIK-IKKα) mediated phosphorylation of *Nfkb2* encoded precursor p100 bound to RelB (*Sun, 2012*). Subsequent proteasomal processing removes the C-terminal inhibitory domain of p100 from RelB:p100 complex to generate RelB:p52 NF-κB dimer, which mediates the expressions of organogenic chemokine genes in the nucleus (*Bonizzi et al., 2004*).

Molecular interaction between the non-canonical signal transducer p100 and RelA has also been charted. In its homo-oligomeric form, termed IκBδ, p100 was shown to utilize its inhibitory domain to sequester a subpopulation of the RelA:p50 dimer (*Basak et al., 2007*; *Savinova et al., 2009*). LTβR through non-canonical NIK-IKK1 signal inactivates IκBδ to induce a weak yet sustained RelA: p50 activity. Conversely, RelA-induced synthesis of p100 and consequent accumulation of inhibitory IκBδ was shown to exert negative feedback limiting canonical RelA activity (*de Wit et al., 1998*; *Legarda-Addison and Ting, 2007*; *Shih et al., 2009*). In addition, an alternate RelA: p52 dimer has been reported which is thought to constitute a minor kappaB DNA binding activity (*Hoffmann et al., 2003*). Crosstalk between apparently distinct cell signaling pathways is known to offer diversity in cellular responses. Despite these connectivities, a plausible role of signal integration via the NF-κB system in regulating inflammatory RelA NF-κB responses has not been investigated.

In a multidisciplinary study combining biochemistry, genetics, and mathematical modeling, here, we characterized a duration code that determines stimulus-specific crosstalk between canonical and non-canonical signaling in fine-tuning inflammatory RelA NF-κB activity. Through such crosstalk, LTβR sustained TLR4 triggered RelA NF-κB responses by supplementing RelA:p52 NF-κB dimer in a positive feedback loop. Finally, we established the physiological significance of crosstalk control of RelA in intestinal epithelial cells (IECs), where, the NF-κB system integrates gut microenvironment derived lymphotoxin signals through *Nfkb2* to calibrate innate immune responses to *Citrobacter rodentium*.

## Results

### A duration code controlling crosstalk between canonical and non-canonical NF-κB signaling

Given the interconnectedness of the canonical and non-canonical arms (see Introduction and *Figure 1A*), we asked if signal integration via the NF-κB system would allow cell-differentiating cues to modulate inflammatory RelA NF-κB responses. Mathematical reconstruction of dynamic networks illuminates emergent properties, such as crosstalk (*Basak et al., 2012*). To explore crosstalk control, we developed a mathematical model, which we termed the NF-κB Systems Model *v*1.0 (Appendix-1), basing on previously published single NF-κB dimer model versions (*Hoffmann et al., 2002*; *Basak et al., 2007*). In our mathematical model, however, we depicted nuclear activation of both the major RelA:p50 dimer and RelA:p52 dimer, which is thought to constitute a minor RelA NF-κB activity. As described in the preceding single dimer models (*Hoffmann et al., 2002*; *Basak et al., 2007*), signal-responsive degradation and resynthesis of IκBα, IκBβ, IκBε, and inhibitory p100/IκBδ dynamically controlled RelA activity. The model was parameterized based on literature, our own measurements (Appendix-1, *Appendix figures 1–5*, *Supplementary files 1–3*), and fitting procedures. Simulating individual TNFR or LTβR regime, we could recapitulate experimentally observed strong, but temporally controlled, activation of RelA NF-κB complexes during canonical IKK2 signaling or the weak induction of RelA dimers during non-canonical NIK-IKK1 signaling, respectively (*Figure 1B*, *Figure 1—figure supplement 1*).

Next, we examined potential crosstalk between IKK2 and NIK-IKK1 inputs in augmenting RelA NF-κB response in silico. To this end, we generated a theoretical library (*Shih et al., 2009*) of 356 kinase activity profiles, where each member possesses distinct peak onset time, peak amplitude, and duration (*Figure 1C*, *Figure 1—figure supplement 2A*). To screen for permissive crosstalk conditions, we fed this library into the model through IKK2 or NIK-IKK1 or both the arms and iteratively simulated respective RelA activities. Then, we computed RelA responses in the co-treatment regime relative to individual cell stimulations to assign crosstalk indexes to different IKK2 and NIK-IKK1 combinations (*Figure 1D*). Plotting the dynamic features of the crosstalk-proficient kinase inputs, we could reveal a critical duration threshold; where IKK2 activities sustained for more than 2 hr were more likely to engage into crosstalk for varied peak amplitudes and inputs with shorter duration were crosstalk inefficient (*Figure 1E* and *Figure 1—figure supplement 2B*). Illustrating a similar but more elaborate duration control, NIK-IKK1 activities longer than 8 hr selectively participated into crosstalk with the canonical pathway.

Inflammatory mediators activate canonical IKK2 with disparate temporal controls. Consistent to the prior report (*Werner et al., 2008*), our kinase assay ('Materials and methods') revealed that IL-1β, an important pro-inflammatory cytokine, only transiently activates IKK2 in mouse embryonic fibroblasts (MEFs) (left panel, *Figure 1F*). In contrast, bacterial LPS through TLR4-induced IKK2 activity that persisted above the basal level even at 24 hr post-stimulation (right panel, *Figure 1F*, *Figure 1—figure supplement 3B*) (*Covert et al., 2005*). Mimicking prolonged signaling during cell-differentiation processes, LTβR engagement using agonistic αLTβR antibody led to sustained activation of the non-canonical NIK-IKK1 (*Figure 1G* and *Figure 1—figure supplement 3A,B*). Using these experimental kinase activities as inputs, our computational simulations revealed insulation of IL-1R signaling from LTβR-mediated crosstalk (left panel, *Figure 1H*), but robust crosstalk between TLR4 and LTβR that amplified late RelA response upon costimulation (right panel, *Figure 1H*). Therefore, our mathematical analyses predicted that a duration code selectively engages long lasting canonical kinase activities into crosstalk with LTβR induced NIK signal to impart stimulus specificity.

### Stimulus-specific crosstalk allows LTβR signal to prolong TLR4 induced RelA NF-κB response

To experimentally verify stimulus specificity of crosstalk control, we measured nuclear RelA activities induced in MEFs by canonical or non-canonical inducers or co-treatment regime that concomitantly activated both the pathways. IL-1R signal, in parallel to transient IKK2 activation, elicited strong RelA activity at 30 min in EMSA that was largely attenuated within 1 hr, whereas, non-canonical LTβR signal only weakly induced RelA and RelB dimers those persisted even at 24 hr (*Figure 2A*). Indeed, we were unable to detect any significant enhancement of RelA activity, relative to IL-1 induced peak, upon

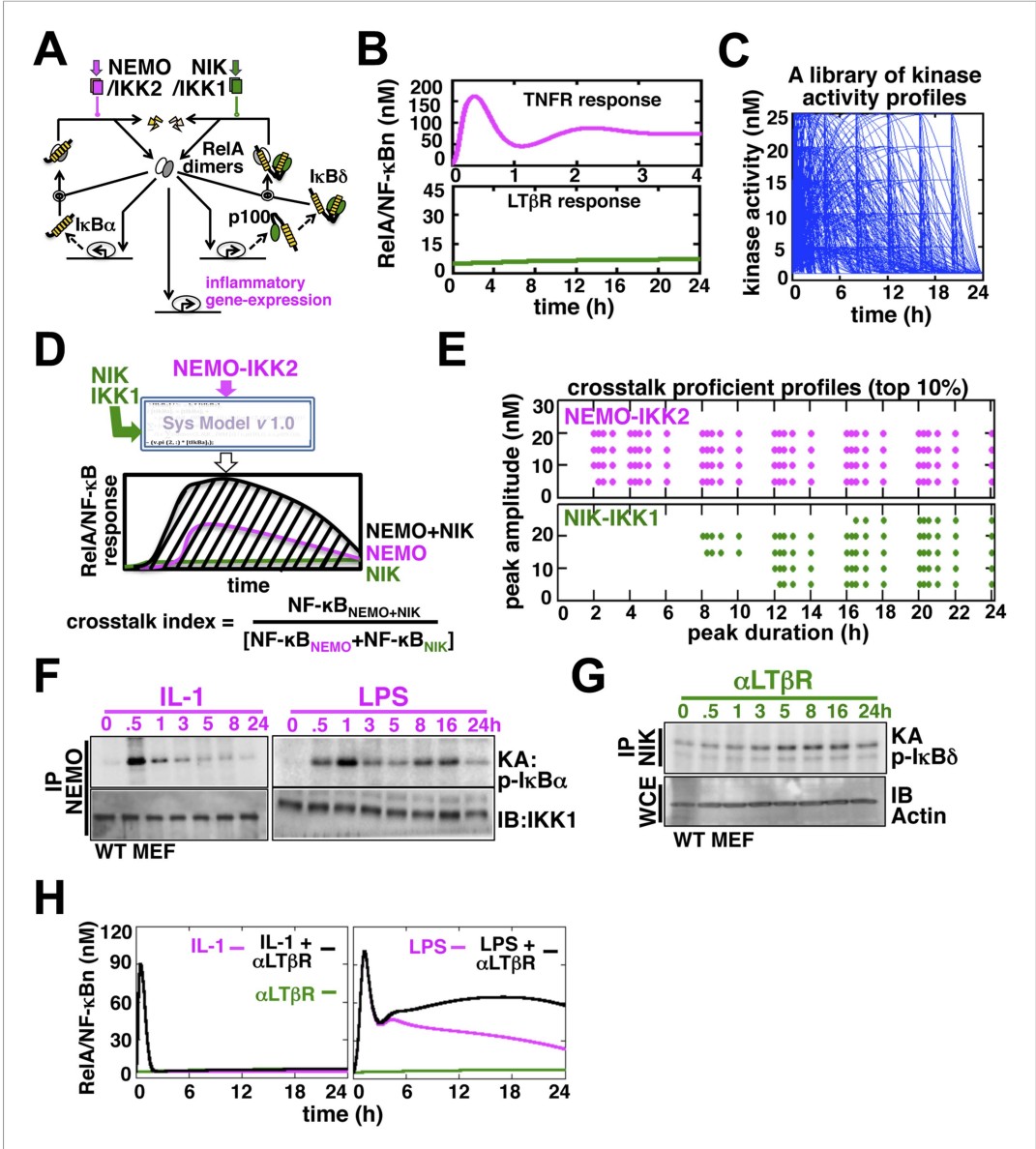

**Figure 1**. Computational simulations predicting a duration code underlying crosstalk control. (**A**) A current model for RelA NF-κB activation via the canonical (IKK2) or the non-canonical (NIK-IKK1) pathways, respectively. RelA dimers represent both RelA:p50 and RelA:p52. Regulation of RelA NF-κB activities through crosstalk between these two pathways has not been addressed. (**B**) Computational simulations of nuclear RelA NF-κB induction by TNFR-induced IKK2 (top, magenta) or LTβR-induced NIK-IKK1 signals (bottom, green). (**C**) A theoretical library of 356 distinct kinase activity profiles. (**D**) A schematic describing in silico crosstalk studies. The kinase inputs were fed into the model through the IKK2 or NIK-IKK1 or both the arms. The RelA NF-κB responses, quantified as baseline corrected total area under the respective activity curves, were used for computing crosstalk indexes. (**E**) Based on their respective crosstalk indexes, top 10% combinations of theoretical IKK2 and NIK1 activity profiles were identified and duration as well as amplitude of the associated crosstalk-proficient IKK2 (top) or NIK-IKK1 (bottom) profiles were plotted. (**F** and **G**) The IKK2 (**F**) or NIK-IKK1 (**G**) activities were monitored by incubating GST-IκBα or GST-IκBδ with NEMO or NIK co-immunoprecipitates derived from MEFs treated with IL-1β, LPS (**F**, left and right panels), or αLTβR (**G**), respectively. For IKK2 assay, co immunoprecipitated IKK1 and for NIK-IKK1 assay, actin present in the cell extracts was used as loading controls. (**H**) Computational simulations predicting augmented RelA activity in LPS+αLTβR (right panel) and a lack of crosstalk in IL-1+αLTβR (left) co-treatment regimes.

The following figure supplements are available for figure 1:

*Figure 1. continued on next page*

*Figure 1. Continued*

**Figure supplement 1**. Analyzing the composition of signal-induced NF-κB dimers.

**Figure supplement 2**. Mathematical modeling revealing a duration code underlying signaling crosstalk.

**Figure supplement 3**. Kinase assays probing cellular activation of NEMO-IKK2 and NIK-IKK1.

---

costimulation (*Figure 2A*). In comparison, canonical TLR4 induced a temporally distinct RelA NF-κB activity with an early peak at 1 hr, subsequent descend and a progressively weakened late phase between 8 hr and 24 hr (*Figure 2B*). Corroborating our mathematical prediction, concomitant LTβR signal sustained NF-κB response triggered by TLR4 (*Figure 2B*). Signal integration via the NF-κB system synergistically enhanced TLR4-induced late RelA activity at 24 hr in the costimulation regime with mostly unaltered RelB response relative to solitary LTβR engagement (*Figure 2B*, quantification and *Figure 2C*). Sequentially engaging MyD88 and Trif, TLR4 was shown to produce extended IKK2 activity (*Covert et al., 2005*). To determine if the observed stimulus specificity of crosstalk is indeed due to the duration of IKK2, we utilized Trif-deficient MEFs that only transiently activated IKK2 upon LPS treatment (*Figure 2D*). Despite a functional non-canonical pathway (*Figure 2—figure supplement 1*), LTβR was restricted from crosstalk with TLR4 in Trif-deficient cells (*Figure 2E*), thereby, suggesting a Trif-dependent mechanism that relies on the duration of canonical IKK2 in imparting stimulus specificity of crosstalk control.

## Signal integration via the NF-κB system amplifies the late expressions of TLR4-induced pro-inflammatory genes

Long-lasting kinase activities are expected to elicit sustained RelA responses. Then what might be the significance of signal integration via the NF-κB system? Interestingly, computational simulations demonstrated only a muted increment in the RelA activity with increasing duration of IKK2 (*Figure 3A*). LTβR induced NIK-IKK1 signal relieved this saturation to fully unravel the NF-κB activation potential of long duration IKK2 signals upon crosstalk. We postulated that the difference in the RelA responses induced by long-lasting IKK2 in the presence or absence of non-canonical signal would decode into differential gene activities.

To evaluate the potential gene-effect of crosstalk, we measured the expressions of several known RelA target chemokine and cytokine genes using quantitative RT-PCR. Our analyses revealed that IL-1 treatment rapidly induces the expressions of TNF, IP-10, and MIP-1α mRNAs within 1 hr with residual expressions at 24 hr post-stimulation (*Figure 3B*). The levels of IL-1β and RANTES mRNAs were insensitive to IL-1 treatment. Also, weak LTβR signal alone did not significantly induce the expressions of these pro-inflammatory genes in MEFs. Indeed, LTβR costimulation was ineffective in augmenting IL-1 induced early or late expressions of the chemokine and cytokine genes (*Figure 3B*). Solitary LPS treatment not only robustly induced the expressions of TNF, IP-10, and MIP-1α mRNAs, but also led to late accumulation of IL-1β and RANTES mRNAs at 24 hr (*Figure 3C*). Consistent to our hypothesis, LTβR costimulation prolonged TLR4-induced gene expressions with further augmented late, but not early, expressions of IL-1β, IP-10, MIP-1α, and RANTES mRNAs. TNF mRNA levels were insensitive to crosstalk regulation (*Figure 3C*).

Furthermore, we used microarray analyses to compare global gene expressions activated by TLR4 or LTβR or both at 24 hr post-stimulation. Estimating normalized crosstalk scores (bottom panel, *Figure 3D*, *Figure 3—source data 1*, 'Materials and methods'), we could reveal a synergistic effect of LTβR on TLR stimulated late gene expressions in WT MEFs. Out of 943 LPS induced genes, however, a select set of 114 genes was further upregulated upon costimulation. Strikingly, gene set enrichment analysis (GSEA) (*Subramanian et al., 2005*) (*Figure 3—source data 2*, 'Materials and methods') demonstrated an enrichment of NF-κB targets among genes positively controlled through crosstalk (middle and top, *Figure 3D*). We have also noted downregulation of several LPS-induced genes in the costimulation regime those appeared less likely to be NF-κB targets in GSEA. Taken together, these analyses substantiated an important function of prolonged RelA activity in crosstalk-amplification of TLR4-induced late expressions of NF-κB target genes, particularly those encode pro-inflammatory chemokines and cytokines.

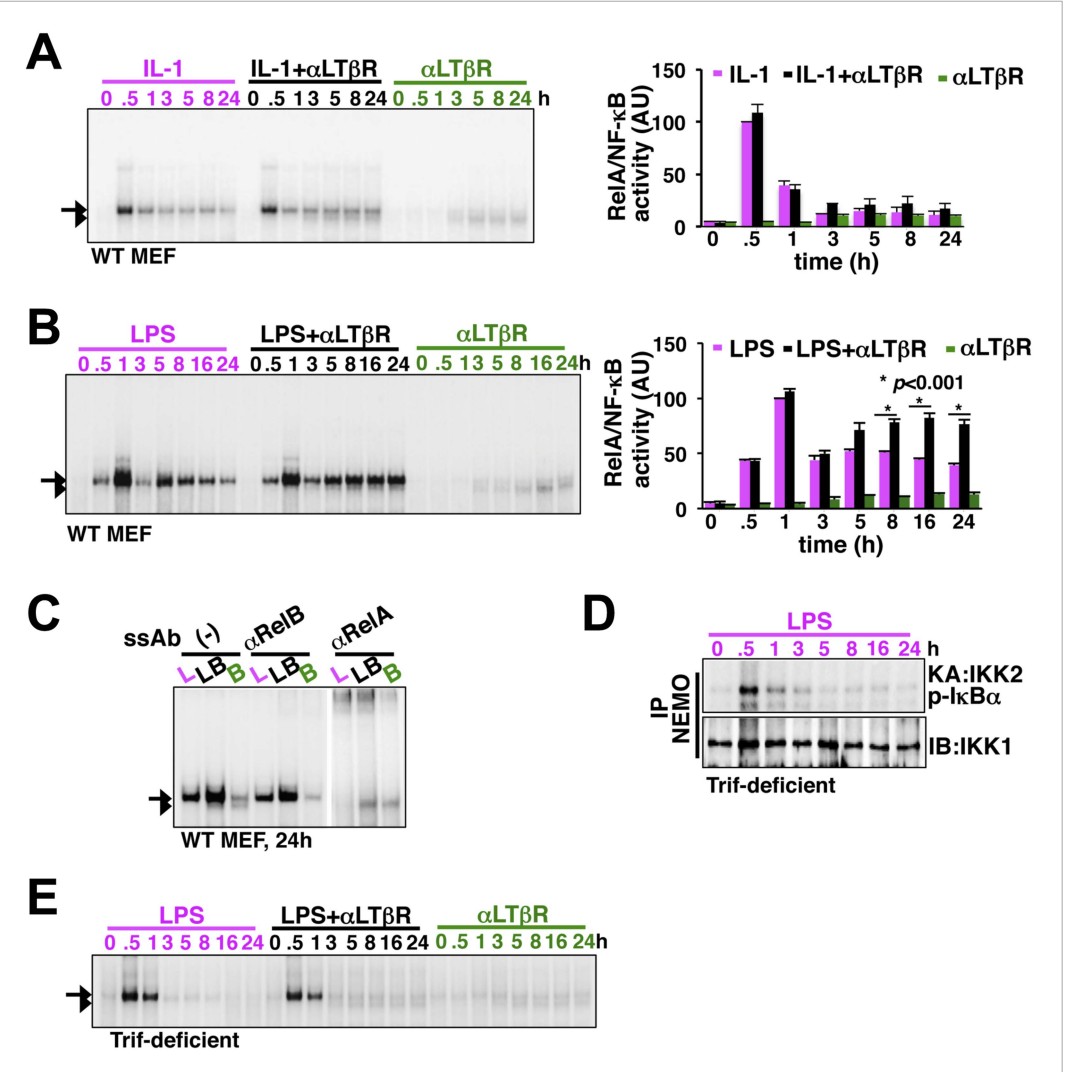

**Figure 2**. LTβR signal sustains TLR4, but not IL-1R, induced RelA NF-κB response. (**A**) Nuclear NF-κB activities induced in MEFs by IL-1β or αLTβR or co-treatment were resolved in EMSA using a κB site containing DNA probe. The faster migrating complex, indicated with an arrowhead, consists of RelB and the slower migrating complex activated by both canonical or non-canonical signaling, denoted with an arrow, consists of RelA dimers. The compositions of the DNA binding complexes were ascertained in *Figure 2C*. Right, signal corresponding to RelA NF-κB activities were quantified and graphed relative to the respective IL-1 induced peak value. Data were expressed as mean of 3 quantified biological replicates ± SEM. (**B**) EMSA result, representative of three independent biological replicates, revealing augmented late NF-κB activities in the co-treatment regime as compared to cell treatment with LPS or αLTβR alone. Right, signal corresponding to RelA NF-κB activities were similarly quantified and graphed relative to LPS induced peak value. Note, late RelA activities in the LPS+αLTβR co-treatment regime were significantly augmented from of the LPS induced activities. (**C**) Supershift analysis distinguishing RelA and RelB dimers induced in MEFs treated with LPS (L) or αLTβR (B) or both (LB) for 24 hr. (**D**) Kinase assay revealing transient IKK2 activities in response to LPS in Trif-deficient MEFs. (**E**) EMSA data, representative of three independent experiments, revealing a lack of NF-κB crosstalk between TLR4 and LTβR in Trif-deficient MEFs.

The following figure supplement is available for figure 2:

**Figure supplement 1**. Analyzing crosstalk control in Trif-deficient MEFs.

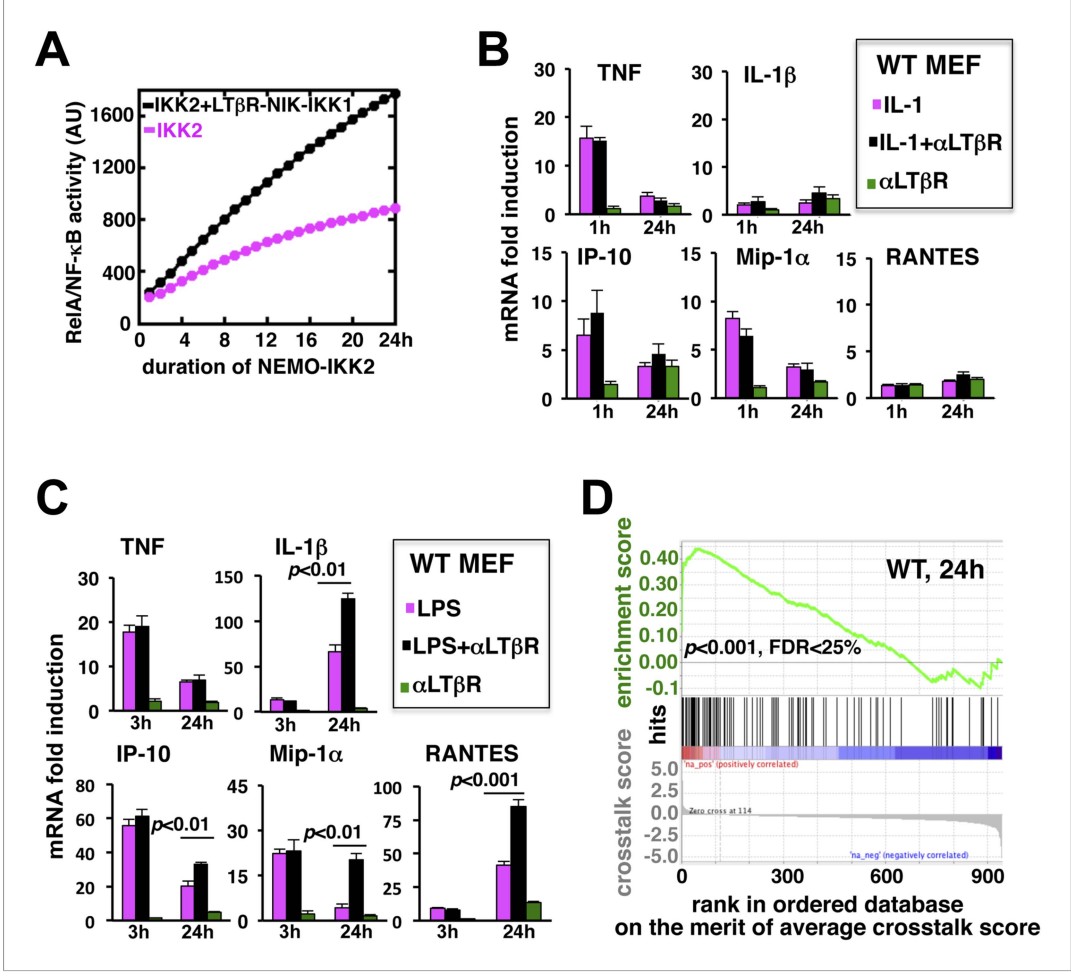

**Figure 3**. LTβR signal augments the late expressions of TLR4-induced NF-κB target genes. (**A**) Computational simulation revealing total RelA activities induced by IKK2 inputs of various durations, in the absence or presence of LTβR induced NIK-IKK1. (**B**) Quantitative RT-PCR measuring early (1 hr) and late (24 hr) expressions of chemokine and cytokine genes in WT MEFs by IL-1R or LTβR or costimulation. (**C**) Gene-expression analyses similarly revealing early (3 hr) and late (24 hr) expressions of chemokine and cytokine genes in WT MEFs by TLR4 or LTβR or costimulation. In (**B**) and (**C**), data are expressed as mean of 3 quantified biological replicates ± SEM. The statistical significance was determined using two-tailed Student's *t*-test. (**D**) LPS-induced genes, identified in representative microarray experiments at 24 hr post-stimulation, were ranked based on their normalized crosstalk score (bottom panels), which reflects synergistic gene activation in the co-treatment regime as compared to individual cell treatments for a positive value. GSEA demonstrated statistically significant enrichment of NF-κB targets (top), with enrichment score of 0.44 for WT MEFs, among genes positively controlled through crosstalk. Hits (middle) indicate NF-κB response genes.

The following source data are available for figure 3:

**Source data 1**. List of LPS target genes positively regulated through crosstalk.

**Source data 2**. A pre-determined list of 290 NF-κB response genes used in GSEA.

## Non-canonical signal transducer *Nfkb2* supplements RelA:p52 dimer to sustain canonical RelA NF-κB responses

To determine the mechanism underlying signal integration via the NF-κB system, we individually perturbed 105 model parameters and quantified relative changes in the crosstalk index ('Materials and methods'). Our parameter sensitivity analysis identified the rate constant associated with the NF-κB-induced transcription of *Nfkb2* as the most critical parameter underlying crosstalk control of RelA

NF-κB activity (*Figure 4A*). As such, *Nfkb2* encodes for both NF-κB inhibitor and NF-κB precursor functions. Computationally simulating individual RelA:p50 and RelA:p52 nuclear activities in the LPS, αLTβR, or costimulation regimes, we could further suggest that the *Nfkb2* precursor function in generating RelA:p52 dimer is important for augmenting RelA NF-κB activity during crosstalk in WT system (*Figure 4B*). Consistently, our modeling analyses predicted complete abrogation of crosstalk in *Nfkb2*⁻/⁻ cells (*Figure 4B*).

Our biochemical studies revealed that LPS induces RelA NF-κB-driven transcription of *Nfkb2* to produce p100 (top panel, *Figure 4C* and *Figure 4—figure supplement 1A*), which was shown to oligomerize as NF-κB inhibitory IκBδ (*Savinova et al., 2009*; *Shih et al., 2009*; *Tao et al., 2014*). Concomitant LTβR signal instead utilized TLR4-induced, newly synthesized p100 to potently generate p52, thereby, neutralizing the inhibitory p100 function (*Figure 4C*). p50 levels were not discernibly altered in these stimulation regimes in our experiments. Our immunoprecipitation based analysis demonstrated that NIK-IKK1 signal relieves RelA from p100/IκBδ-mediated inhibition during crosstalk (*Figure 4D*). Intriguingly, LTβR costimulation of MEFs for 24 hr also produced ~fourfold more RelA:p52 NF-κB dimer as compared to solitary LPS treatments (*Figure 4D*). Limited transcriptional up-regulation of *Nfkb2* by weak LTβR signal was correlated with only the modest p52 and RelA:p52 generation (*Figure 4C,D*). In supershift assay, we could ascertain that RelA:p52 dimer generated upon LTβR costimulation appears as a strong nuclear DNA binding activity at 24 hr (*Figure 4E*) to supplement to the TLR4-induced RelA NF-κB responses. An important role of dimerization in stabilizing NF-κB monomers from degradation has been reported earlier (*Fusco et al., 2008*). Interestingly, RelA protein rapidly accumulated in cells upon proteasome inhibition (*Figure 4F*) suggesting a robust constitutive degradation mechanism that offsets basal synthesis of RelA monomers in maintaining the steady-state level. Our study indicated that this enduring flux ensures copious supply of RelA to bind to de novo generated p52, produced from the newly synthesized p100 during crosstalk.

Next, our genetic analyses revealed that canonical RelA:p50 response to TLR4 signal, primarily controlled through classical IκBs, is largely intact in *Nfkb2*⁻/⁻ with early induction and diminished late activities comparable to WT MEFs (*Figure 4G* and *Figure 4—figure supplement 1B*). Consistent to the prediction based on computational modeling studies, a lack of RelA:p52 dimer generation in *Nfkb2*⁻/⁻ cells, however, ablated LTβR-mediated enhancement of TLR4-induced late RelA DNA binding activity (*Figure 4G*) as well as crosstalk amplification of RelA target pro-inflammatory gene expressions (bottom panel, *Figure 4H*). LTβR costimulation not only enhanced TLR4 induced RelA DNA binding but also activated RelB dimers through the non-canonical pathway. Prior reports have indicated cell-type specific inhibitory as well as activating role of RelB in chemokine gene expressions (*Weih et al., 1996*; *Shih et al., 2012*). Importantly, costimulation of *Relb*⁻/⁻ MEFs led to similar hyperactivation of LPS-induced late expressions of IL-1β, IP-10, and RANTES mRNAs as in WT cells (compare top panel, *Figure 4H* with *Figure 3C*). Although, the crosstalk effect on MIP-1α expressions was somewhat muted owing to prolonged expression of this gene in *Relb*⁻/⁻ MEFs in response to solitary LPS treatment. Therefore, our analyses confirmed that the precursor function encoded by *Nfkb2* in generating RelA:p52 NF-κB dimer is critical for integrating lymphotoxin derived signals to the pro-inflammatory RelA NF-κB pathway. Our analyses also suggested that LTβR costimulation led to the hyperactivation of LPS-induced expressions of chemokine and cytokine genes in an *Nfkb2*-dependent manner with only a minor, if any, role for *Relb*.

## Inducible synthesis of *Nfkb2* by canonical signal triggers a positive feedback loop during crosstalk

Given the computational prediction for an important role of NF-κB-induced transcription of *Nfkb2*, we compared the inducible expression of the crosstalk mediator *Nfkb2* in response to LPS or IL-1 to understand the molecular basis of stimulus-specific control. As opposed to rapid expression of *Nfkbia* mRNA, which encodes IκBα, LPS induced *Nfkb2* mRNA with a delay (top panel, *Figure 5A*). Similarly, chronic TNF treatment induced *Nfkb2* mRNA in WT MEFs with an explicit 1 hr delay (*Figure 5B*) that was also observed earlier and incorporated in both the previous (*Basak et al., 2007*) as well as the current mathematical model versions. Analogous time lags were observed in the expression of several inflammatory genes those require additional chromatin modifications for the initiation of RelA-induced transcription (*Natoli et al., 2005*). When *Nfkb2* transgene was stably expressed in *Nfkb2*⁻/⁻ MEFs from an exogenous NF-κB responsive promoter, *Nfkb2* mRNA was readily induced by TNF without a delay (*Figure 5B*). Remarkably, IL-1 treatment was ineffective in activating the expression of *Nfkb2* mRNA in

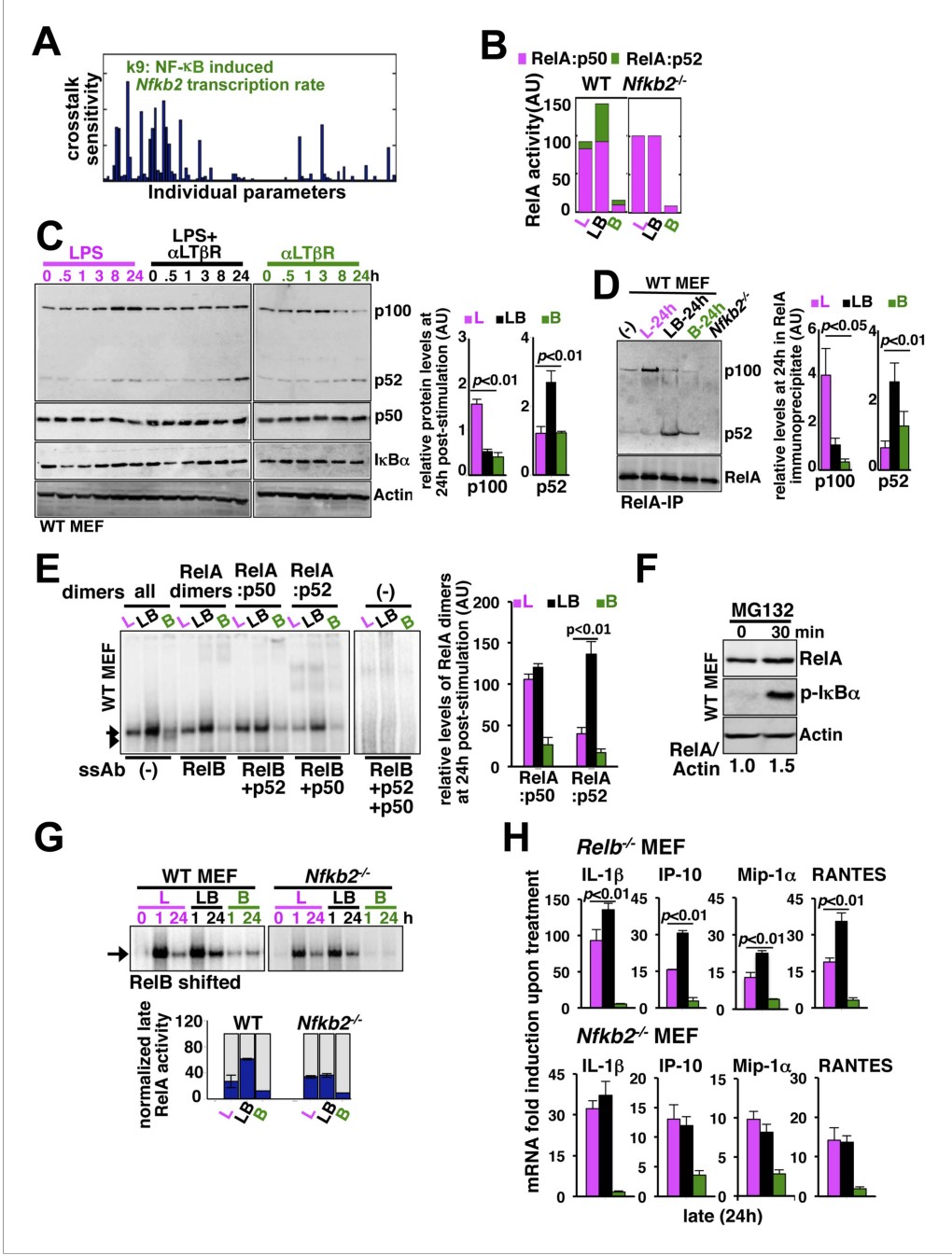

**Figure 4**. Signal generation of RelA:p52 NF-κB dimer underlies a pro-synergistic function of *Nfkb2*. (**A**) Local sensitivity analysis revealing the effect of perturbation of the individual biochemical parameters on crosstalk between TLR4 and LTβR. (**B**) Computational simulations of total RelA:p50 and RelA:p52 activities between 8 and 24 hr in response to LPS, αLTβR, or both in WT or *Nfkb2* deficient systems. (**C**) Immunoblot charting cellular abundance of NF-κB/IκB proteins during signaling. Right, signal corresponding to p100 and p52 levels at 24 hr post-stimulation were quantified and graphed. (**D**) Immunoblot of RelA co-immunoprecipitates, normalized for the RelA content, obtained using whole cells extracts derived from MEFs treated with LPS (L) or αLTβR (B) or both (LB) for 24 hr. The quantified data demonstrates the level of RelA-p100 or RelA-p52 complexes at 24 hr post-stimulation. (**E**) Supershift analysis revealing the composition of the RelA dimers induced upon indicated cell treatments for 24 hr. Right, signal corresponding to RelA:p50 or RelA:p52 NF-κB activities were quantified and graphed. (**F**) Representative immunoblot demonstrating an increase in the RelA protein level in MEFs, in parallel to phospho-IκBα accumulation, upon proteasome inhibition using MG-132. (**G**) EMSA revealing RelA activities induced in WT or *Nfkb2⁻/⁻* MEFs upon indicated cell-stimulations by supershifting RelB. Below, quantified late (24 hr) RelA activities were plotted for

*Figure 4. continued on next page*

*Figure 4. Continued*

different genotypes subsequent to normalizing against the respective LPS induced early 1 hr peak. (**H**) mRNA analyses comparing late (24 hr) expressions of chemokines/cytokines in *Relb*⁻/⁻ (top) and *Nfkb2*⁻/⁻ (bottom) MEFs upon indicated cell stimulations. Quantified data for both biochemical and gene-expression analyses presented in this figure are expressed as mean of 3 biological replicates ± SEM.

The following figure supplement is available for figure 4:

**Figure supplement 1**. An important role of the non-canonical signal transducer *Nfkb2* in crosstalk.

---

WT MEFs, despite the early induction of *Nfkbia* (bottom, *Figure 5A*). Only upon eliminating the transcriptional delay, our mathematical model could simulate *Nfkb2* mRNA induction by IL-1 treatment (*Figure 5C*). Indeed, we could also experimentally rescue the defect in *Nfkb2* mRNA induction by IL-1 treatment in the engineered *Nfkb2*⁻/⁻ cell line, which expresses *Nfkb2* transgene from the NF-κB-responsive promoter without the delay (*Figure 5D*). Consistent to our computational identification that NF-κB inducible transcription of *Nfkb2* is important, disruption of NF-κB-inducible synthesis by expressing p100 from a constitutive promoter in *Nfkb2*⁻/⁻ MEFs abrogated crosstalk amplification of TLR4-induced late NF-κB activity by concomitant LTβR signal (*Figure 5E* and *Figure 5—figure supplement 1A*). While the NF-κB-responsive expression of *Nfkb2* transgene in *Nfkb2*⁻/⁻ cells restored the crosstalk effect at the level of RelA NF-κB activation (*Figure 5E*) by potentiating RelA:p52 induction in LPS+αLTβR costimulation regime (*Figure 5F*, *Figure 5—figure supplement 1B*).

Our results confirmed that signal induction of *Nfkb2* is important for crosstalk and suggested that a promoter intrinsic delay necessitates persistent canonical signal for RelA-mediated induction of pro-synergistic *Nfkb2*. Such delay encoding insulated IL-1R signaling, which transiently activates IKK2 and RelA by restricting *Nfkb2* mRNA expressions, and accounted for the abrogated crosstalk in Trif-deficient MEFs that transiently activated the NF-κB pathway. Our studies also explained the requirement for the long-duration NIK-IKK1 signals in targeting this late acting p100 *Nfkb2* feedback for RelA:p52 dimer generation. In contrast, IL-1 signal led to early induction of *Nfkb2* mRNA expressions in the engineered cells, which inducibly express *Nfkb2* transgene without the delay (*Figure 5D*). Pretreatment of these engineered cells with αLTβR for 8 hr and subsequent IL-1 stimulation that effectively converged the non-canonical signal to IL-1-induced *Nfkb2* feedback, potentiated p52 production (*Figure 5—figure supplement 1C*) and prolonged IL-1-induced RelA response (*Figure 5G* and *Figure 5—figure supplement 1D*). Correlating with the early onset of *Nfkb2* mRNA induction in response to IL-1 treatment, observed crosstalk effects were indeed obvious within 1 hr of IL-1 treatment in these engineered cells.

In sum, we elucidate a crosstalk mechanism that discriminates between TLR4 engagement and concomitant cell activation through TLR4 and LTβR (*Figure 5H*). Negative feedbacks by IκBα and p100/IκBδ coordinately terminate canonical TLR4 response. But, *Nfkb2* functions pro-synergistically upon costimulation; in a positive feedback loop, non-canonical LTβR signal targets the newly synthesized p100, abundantly produced by TLR4, to potently generate p52 and RelA:p52 dimers in sustaining inflammatory RelA NF-κB responses. Importantly, RelA:p50 and RelA:p52 heterodimers were shown to share DNA binding and gene-expression specificities (*Siggers et al., 2012*; *Zhao et al., 2014*). Our experimental data also indicated that RelA:p52 dimer has comparable efficiency in inducing the expression of *Nfkb2* mRNA as the RelA:p50 dimer (Appendix-1, *Appendix figure 4C*). Although, emergent crosstalk is expected to be controlled by several biochemical constrains, the transcriptional delay intrinsic to the *Nfkb2* promoter appears to be critical for the duration code and thereby the stimulus specificity.

## *Nfkb2* integrates lymphotoxin signal within intestinal niche to reinforce epithelial NF-κB responses to *C. rodentium*

In addition to its role in lymph node development during embryogenesis, recent studies have illustrated a requirement for LTβR in innate immune responses in adult mice. Disruption of LTβR signal using LTβR-Ig fusion protein was shown to compromise innate immune responses upon subsequent infection with *C. rodentium*, a natural mouse enteric pathogen that led to mortality (*Spahn et al., 2004*; *Wang et al., 2010*). IEC-specific deletion of LTβR similarly obliterated bacterial clearance (*Wang et al., 2010*). The engagement of ligand-expressing RORγt⁺ innate lymphoid cell is thought to

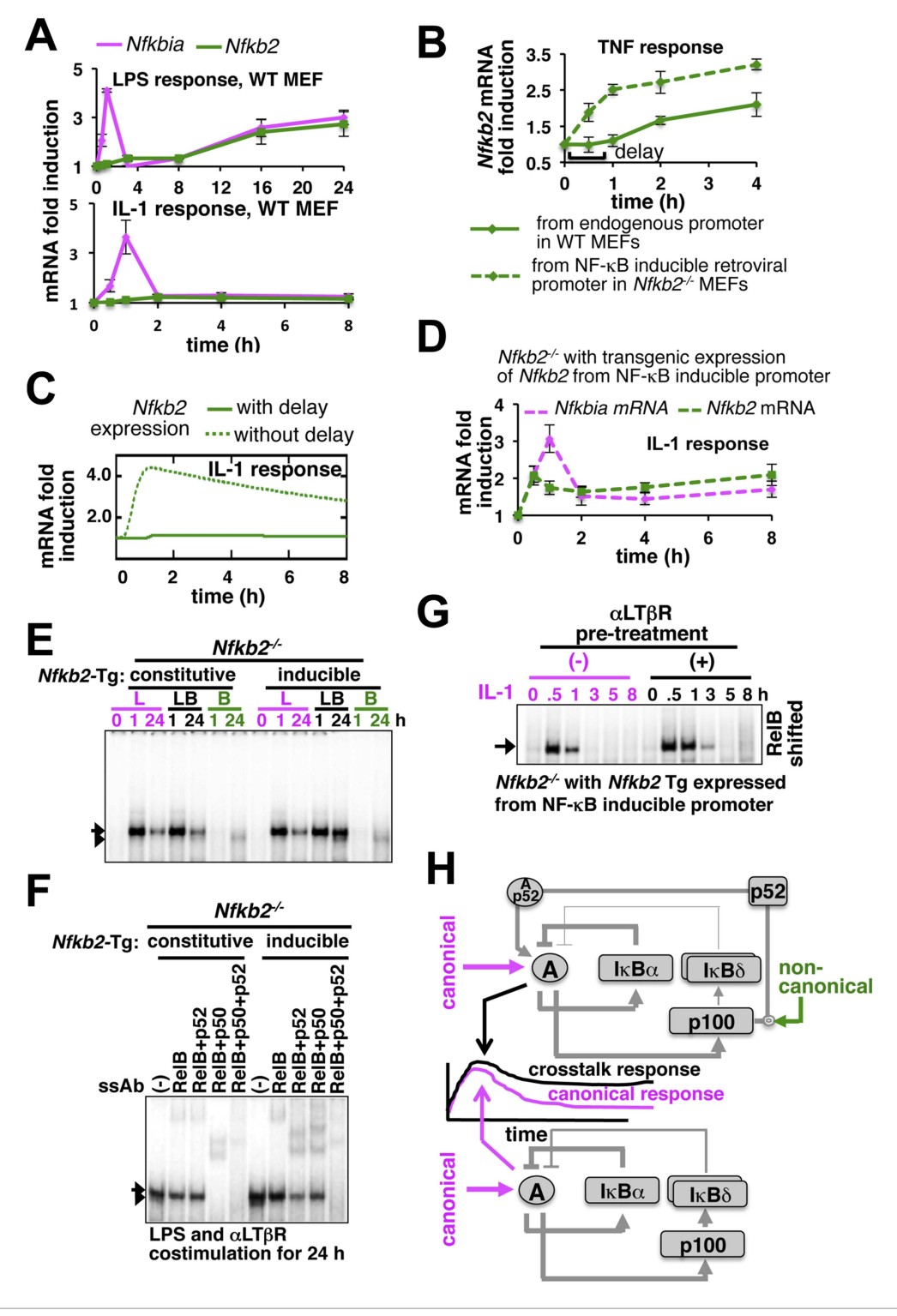

**Figure 5**. Induction of *Nfkb2* expressions by canonical signal is required for crosstalk. (**A**) Relative levels of *Nfkb2* mRNA and *Nfkbia* mRNA, which encodes IκBα, in WT MEFs during LPS or IL-1 signaling. (**B**) TNF induced delayed expression of *Nfkb2* mRNA in WT MEFs and rapid production in an engineered *Nfkb2⁻/⁻* cell-line from an exogenous NF-κB dependent promoter. Data presented in (**A**) and (**B**) are expressed as mean of 3 quantified biological replicates ± SEM. (**C**) Simulations comparing IL-1-induced *Nfkb2* mRNA expressions in the presence or absence of transcriptional delay. (**D**) Quantitative RT-PCR revealing IL-1 induced expression of *Nfkb2* and *Nfkbia*
*Figure 5. continued on next page*

*Figure 5. Continued*

mRNAs in the engineered *Nfkb2*<sup>−/−</sup> cell line with transgenic expressions of *Nfkb2* from the NF-κB inducible promoter. Data are expressed as mean of 3 quantified biological replicates ± SEM. (**E**) EMSA comparing NF-κB activities induced in *Nfkb2*<sup>−/−</sup> MEFs expressing *Nfkb2* from either a constitutive (lane 1–7) or an NF-κB responsive (lane 2–14) transgenic (Tg) promoter. (**F**) Supershift analyses comparing nuclear abundance of different NF-κB dimers activated upon costimulation with LPS+αLTβR for 24 hr in these two engineered *Nfkb2*<sup>−/−</sup> cell lines. (**G**) The engineered *Nfkb2*<sup>−/−</sup> cell line, which expresses *Nfkb2* from the NF-κB-inducible promoter, was pretreated for 8 hr with αLTβR and subsequently treated with IL-1. Supershifting RelB, representative RelA activities were captured in EMSA. (**H**) A graphical depiction of the proposed crosstalk control; two negative feedback loops coordinately attenuate TLR4 responses. However, one of these negative feedback loops is converted into a positive feedback loop by non-canonical signals to generate crosstalk at the level of RelA NF-κB activation. Magenta and green arrows indicate canonical IKK2 and non-canonical NIK-IKK1 inputs, respectively, and line thickness signifies relative strength of feedbacks.

The following figure supplement is available for figure 5:

**Figure supplement 1**. Induction of *Nfkb2* expression by canonical signal is important for crosstalk.

---

provide the critical lymphotoxin signal in colon during the course of bacterial infection (*Upadhyay and Fu, 2013*). A requirement of epithelial RelA activity in the chemokine gene expressions has been documented earlier (*Alcamo et al., 2001*). Given our identification of a costimulatory function of LTβR in inflammatory RelA activation, we asked if signal integration via the NF-κB system could explain the epithelial requirement of LTβR in innate immune responses in vivo.

First, a functional non-canonical pathway downstream of LTβR (*Figure 6—figure supplement 1A*) augmented the RelA activity induced by pathogen sensing TLR4 in otherwise hypo-responsive MSIE colon epithelial cell-line (*Figure 6A*). Next, we biochemically analyzed NF-κB activation in IECs derived from WT mice intraperitoneally injected with antagonistic LTβR-Ig or a control-Ig 1 day prior to oral infection with *C. rodentium*. Upon colonization, *Citrobacter* initially triggered epithelial accumulation of p100 that was fully processed into p52 by day5 (*Figure 6B*) generating RelA:p52 dimer (*Figure 6C*) in control-Ig, but not LTβR-Ig, treated mice. Bacterial infection elicited RelA DNA binding activity in IECs that gradually accumulated in the nucleus (*Figure 6D,E*) with substantial contribution from RelA:p52 dimer along with RelA:p50 dimer at day5 post-infection (*Figure 6—figure supplement 1B*). Our supershift analyses further confirmed complete absence of RelB containing NF-κB DNA binding activity in IECs derived from infected mice (*Figure 6E*). Perturbing LTβR signal attenuated RelA NF-κB activation with more obvious defects at day5 (*Figure 6D*). Likewise, pathogen-responsive RelA activation in IECs derived from *Nfkb2*<sup>−/−</sup> mice was severely weakened at day5 (*Figure 6F*) that led to significantly reduced expressions of the RelA target chemokines encoding KC and MIP-2α as compared to WT mice (*Figure 6G*). Indeed, infected *Nfkb2*<sup>−/−</sup> mice exhibited diminished neutrophil recruitment in the lamina propria, as revealed by anti-myeloperoxidase immunostaining of the colon sections (*Figure 6H*). Sustained epithelial RelA activity that relies on LTβR mediated processing of pathogen-induced p100 into p52, therefore, mirrored our MEF-based analyses depicting crosstalk between canonical and non-canonical signaling. Collectively, our results connected the previously reported epithelial requirement of LTβR (*Wang et al., 2010*) and NIK (*Shui et al., 2012*) in innate immune response to the NF-κB system in reinforcing RelA activity through *Nfkb2* mediated crosstalk control. Subdued epithelial NF-κB activation, and not hyper-induction, in IECs from infected *Nfkb2*<sup>−/−</sup> mice also suggested that a dominant precursor function of p100 supplying RelA:p52 dimer prolongs RelA response within the intestinal niche.

## Stromal expression of *Nfkb2* is required for limiting *C. rodentium* infection

While WT mice efficiently eliminated infections, increased fecal excretion of bacteria at day10 post-infection in *Nfkb2*<sup>−/−</sup> mice (*Figure 7A*) indicated an inadequacy in limiting local infection, thereby, correlating with the observed defects in the early innate inflammatory response in this knockout (*Figure 6*). Histological analysis of the shrunken colons (*Figure 7—figure supplement 1A,B*) derived from the infected *Nfkb2*<sup>−/−</sup> mice further revealed exacerbated damage with signatures of submucosal leukocyte infiltration (*Figure 7B*). Breach in the intestinal barrier was accompanied by systemic

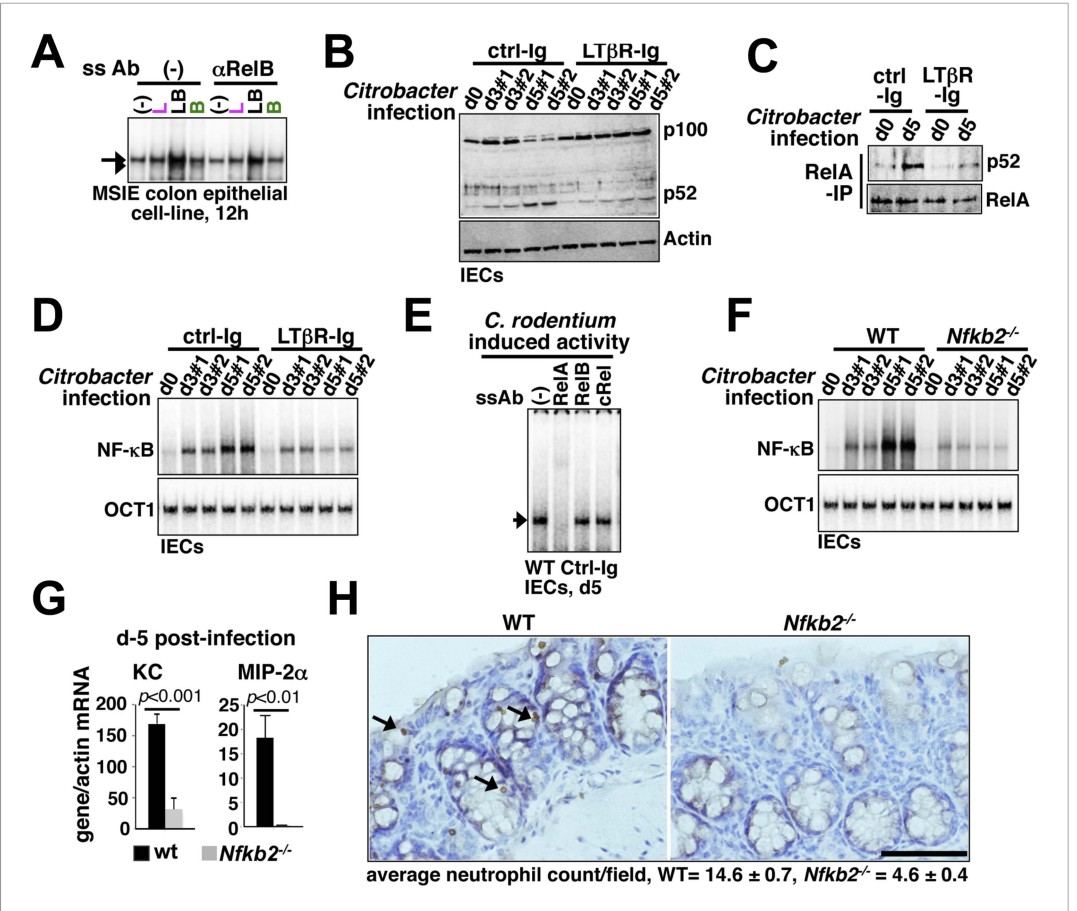

**Figure 6**. *Nfkb2* dependent LTβR crosstalk prolongs RelA NF-κB response in the colon of *Citrobacter rodentium*-infected mice. (**A**) EMSA data, representing two independent experiments, revealing LPS-induced total RelA NF-κB activities induced in MSIE colon epithelial cell line at 12 hr in the absence or presence of 1 μg/ml of αLTβR. (**B**), (**C**), and (**D**) WT mice were injected with control-Ig or LTβR-Ig (n = 2) 1 day prior to infection with *C. rodentium*. IECs were isolated at day3 and day5 post-infection and analyzed for p52 and p100 levels by immunoblotting (**B**), RelA:p52 complex formation by immunoprecipitation-based assay (**C**) or NF-κB DNA binding activities in EMSA (**D**). OCT1 DNA binding activity served as loading control. (**E**) Supershift analyses revealing that exclusively RelA/NF-κB dimer are activated in IECs derived from mice infected with *C. rodentium*. (**F**) NF-κB activities induced in IECs derived from infected WT and *Nfkb2⁻/⁻* mice (n = 2) were similarly measured. (**G**) Epithelial expressions of KC and MIP-2a mRNA derived from WT and *Nfkb2⁻/⁻* mice (n = 5) at day5 post-infection. Data are expressed as mean of 3 quantified biological replicates ± SEM. The statistical significance was determined using two-tailed Student's *t*-test. (**H**) Representative data showing antimyeloperoxidase staining of neutrophils in colons of WT and *Nfkb2⁻/⁻* mice at day4 post-infection. Colon sections from three animals per set and five fields/section were used for quantification and presented as mean ± SEM. The panels with 40× magnification have been presented using scale bars that represent 200 μm.

The following figure supplement is available for figure 6:

**Figure supplement 1**. Control of epithelial RelA NF-κB activation through signaling crosstalk.

bacterial dissemination with increased count in blood (left panel, *Figure 7C*) and liver (right, *Figure 7C*). Finally, bacterial colitis induced in *Nfkb2⁻/⁻* mice resulted in significant body weight loss (*Figure 7D*) and onset of mortality as early as day10 post-infection (*Figure 7E*). Next, we performed reciprocal bone marrow transfer experiments between WT and *Nfkb2⁻/⁻* mice to ensure that the observed sensitivity was not due to previously reported hematopoietic defects in *Nfkb2⁻/⁻* mice (*Caamaño et al., 1998*). WT bone marrow cells in *Nfkb2⁻/⁻* recipients (*Figure 7—figure supplement 1C*) were unable to prevent the infection-related colon pathology (*Figure 7F*), reductions in the

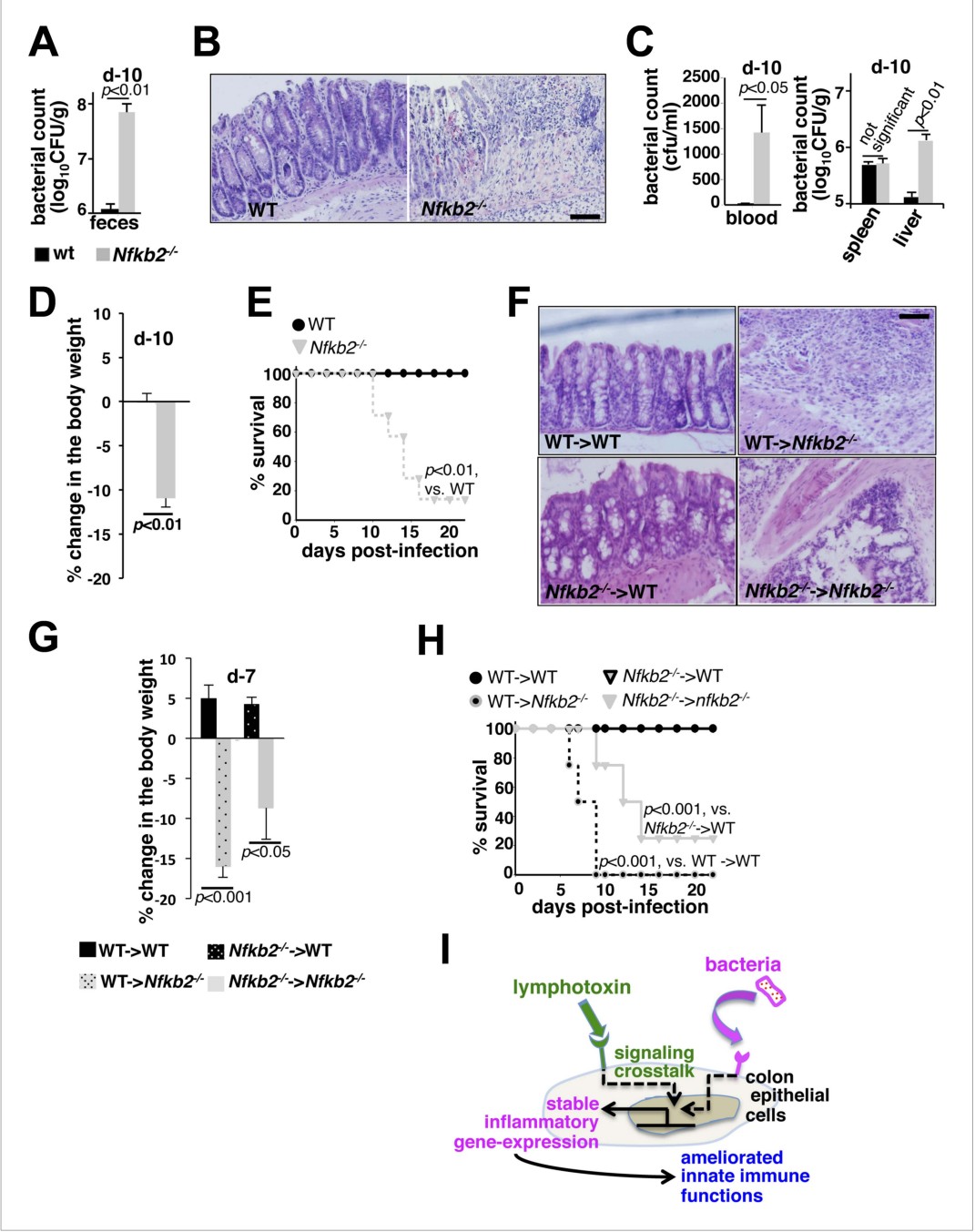

**Figure 7**. A protective role of pro-synergistic *Nfkb2* in the non-hematopoietic compartment to *Citrobacter* infection. (**A**) and (**C**) Bacterial titers in the fecal homogenate (**A**), blood (**C**, left panel) or spleen and liver homogenate (**C**, right) derived from WT and *Nfkb2−/−* mice (n = 4) at day10 post-infection. Data are expressed as mean ± SEM. The statistical significance was determined using two-tailed Student's *t*-test. (**B**) and (**F**) H&E staining of the representative colon sections derived from WT and *Nfkb2−/−* mice at day10 (n = 4, five fields/section) (**B**) or indicated bone marrow chimeras at day7 (n = 2, five fields/section) (**F**) after inoculation. The panel shows 20× magnification with scale bars representing 200 μm. (**D**) and (**G**) Average change in the body weight of WT and *Nfkb2−/−* mice (n = 7) (**D**) or indicated bone marrow chimeras (n = 4) (**G**) upon infection. Data are expressed as mean ± SEM. The statistical significance was determined using two-tailed Student's *t*-test. (**E**) and (**H**) Survival rates of WT and *Nfkb2−/−* mice (n = 7) (**E**) or indicated bone marrow chimeras (n = 4) (**H**) infected with *C. rodentium*. The statistical
*Figure 7. continued on next page*

*Figure 7. Continued*
significance was determined using log rank (Mantel–Cox test). (I) A model depicting the proposed regulatory role of crosstalk in sustaining colonic inflammatory immune responses.
The following figure supplement is available for figure 7:

**Figure supplement 1**. *Nfkb2* controls gut immune response to *Citrobacter* infection.

body weight (*Figure 7G*) and morality (*Figure 7H*). In contrast, WT recipients receiving either WT or *Nfkb2*⁻/⁻ bone marrow resolved infections with comparable efficiencies (*Figure 7F–H*).

In addition to modulating RelA activity through crosstalk, *Nfkb2* also mediates RelB:p52 activation in response to singular non-canonical stimuli. Heightened RelB:p52 activation in B-cells was shown to strengthen host defense in *Otu7b*⁻/⁻ mice (*Hu et al., 2013*). Our inability to rescue the infection-related mortality in *Nfkb2*⁻/⁻ mice using WT hematopoietic cells indicated that the protective function of *Nfkb2* lies within stromal cells. Despite the presence of mesenteric lymph nodes (*Lo et al., 2006*), additional stromal defects in Peyer's patches in *Nfkb2*⁻/⁻ mice (*Yilmaz et al., 2003*) could impair mucosal IgA responses. Although IgM levels were comparable, very low IgA levels at day7, prior to the onset of mortality in *Nfkb2*⁻/⁻ recipients, precluded bona fide comparisons in the radiation chimeras. Interestingly, adult mice, similar to those utilized in our study, lacking IgA or its transporter pIgR efficiently resolved *Citrobacter* infections (*Maaser et al., 2004*). μMT mice depleted of peripheral B-cells presented few mucosal changes in the first 2 weeks of infection and were completely free of mortality (*Simmons et al., 2003*). Finally, *Rag1*⁻/⁻ mice succumbed to *Citrobacter* only late during infections in the 4th week (*Wang et al., 2010*). Though, our study does not rule out crosstalk independent engagement of the *Nfkb2* pathway in activating RelB:p52 dimer in the hematopoietic compartment at a later stage of infection, epithelial RelA NF-κB activation defects coupled to aggravated early colon pathology and early onset of mortality in *Nfkb2*⁻/⁻ mice suggested that the stromal requirement of *Nfkb2*, at least in part, lies within the intestinal epithelial cells in the initial events controlling early innate immunity and involves crosstalk regulation of RelA NF-κB activity (*Figure 7I*, also discussed later).

## Discussion

The NF-κB system transduces signals from a variety of cell-activating stimuli. Our study suggested that such pleiotropic system enables tuning of cellular response to an instructive signal by microenvironment-derived additional costimulatory signals. However, a duration code selectively integrates costimulatory LTβR signal with the TLR4 pathway, insulating transient cytokine signaling, secondary to microbial infections, from crosstalk amplifications. Recent studies have identified important positive feedback regulations underlying dose threshold control of NF-κB response in B-cells (*Shinohara et al., 2014*) or in myeloid cells (*Sung et al., 2014*). Our crosstalk study illuminated a role of a positive feedback loop in sustaining NF-κB response; where non-canonical LTβR signal prolonged canonical TLR4 response by targeting RelA induced p100 for the generation of RelA:p52 NF-κB dimer. Indeed, a dominant precursor function of p100 in producing RelA:p52 led to ablated crosstalk in *Nfkb2*⁻/⁻ cells. As such, non-canonical signal generates RelB:p52 by removing the C-terminal domain of p100 present in the preexisting RelB:p100 dimeric complex (*Sun, 2012*) or liberates RelA:p50 dimer from the p100/IκBδ-inhibited complexes (*Basak et al., 2007*). Signal generation of the RelA:p52 dimer requires both canonical induction of p100/*Nfkb2* expressions and concomitant processing of the newly synthesized p100 into p52 by non-canonical signal. Previous report demonstrating that p100 could be efficiently processed cotranslationally (*Mordmüller et al., 2003*) explained the requirement of canonical signals in amply synthesizing nascent p100 as a substrate for the abundant production of p52 subunit during crosstalk. More so, mature p52 could readily dimerize with RelA, despite a preference of full-length p100 for RelB binding (*Fusco et al., 2008*). These observations along with our current study, thereby, elaborated a requirement of convergence of canonical and non-canonical signals in synergistically generating and activating RelA:p52 dimer. Nevertheless, consistent to the observed gene-effect of crosstalk in our study, a significant overlap between RelA:p50 and RelA:p52 dimers in DNA binding (*Siggers et al., 2012*) and pro-inflammatory gene expressions (*Hoffmann et al., 2003*) was reported earlier. Of note, our results rely on bulk measurements of signaling intermediates and deterministic modeling approaches. Given

studies documenting cell-to-cell variations in signal-induced NF-κB responses (*Lee et al., 2009*), it would be interesting to examine the potential implication of signal integration at the single cell level in determining cellular heterogeneity.

TLR4 activation of epithelial RelA was implicated in the chemokine gene expressions and neutrophil recruitment upon bacterial infections (*Khan et al., 2006*). Yet, epithelial LTβR (*Wang et al., 2010*) was also important for effective innate immune responses to *Citrobacter*. In our proposed model (*Figure 7I*), we could clarify that LTβR provides a critical costimulatory signal through *Nfkb2* to sustain RelA NF-κB response to pathogens in otherwise hyporesponsive colonic epithelial cells. Such signal integration ameliorated innate immune functions by enhancing pro-inflammatory gene expressions. Interestingly, p100⁻/⁻ mice, which lacked the expression of p100, but aberrantly produced p52, revealed hyperplasia of gastric epithelial cells and elevated expressions of RelA target genes (*Ishikawa et al., 1997*). In *Nlrp12*⁻/⁻ mice, robust p52 generation in stromal cells through the non-canonical pathway led to colon cancer associated inflammation (*Allen et al., 2012*), a hallmark for aberrant RelA activity. These studies indeed support a possible role of *Nfkb2* in mucosal epithelial cells in strengthening RelA activity. LTβR engagement in the dendritic cells within colonic patches was shown to trigger IL-22 production by innate lymphoid cells involving cell–cell communications to potentiate gut immunity (*Tumanov et al., 2011*). Although, a defect in the colonic patches in *Nfkb2*⁻/⁻ mice could impair IL-22 mediated protective responses, our cell-intrinsic crosstalk model explained that the reported epithelial requirement of LTβR (*Wang et al., 2010*) is in sustaining RelA NF-κB response during bacterial infection. In future, tissue-specific knockouts may help to further distinguish between innate immune functions of *Nfkb2* in different cell types.

The underlying mechanism and biological functions of RelB:p52 dimer activated by non-canonical signal is well established. From the perspective of signaling crosstalk, our study offers a significant revision of our understanding of non-canonical signaling in amplifying canonical RelA responses. It extends the intriguing possibility that cell-differentiating cues present in the tissue microenvironment may play a more direct role, separate from merely determining the differentiation states of the resident cells, in calibrating innate immune responses by engaging into cell-autonomous signaling crosstalks. Future studies ought to further examine the signal integration via *Nfkb2* in potentiating immune responses against other microbial pathogens. More so, the potential involvement of the deregulated crosstalk control in the pathophysiology of inflammatory disorders, particularly those involving gut, remains to be addressed.

## Materials and methods

### Mice, cells and recombinant DNA

Wild-type or gene-deficient C57BL/6 mice were housed at NII small animal facility and used in accordance with the IAEC guidelines. Primary MEFs were generated from E12.5–14.5 embryos. Late passage Trif-deficient and NIK-deficient MEFs have been described (*Basak et al., 2007*). MSIE cell line was a gift from R. Whitehead, Ludwig Cancer Research. Mouse *Nfkb2* was stably expressed from a promoter containing five tandem kappaB sites from HRS.puro or from a constitutive promoter from pBabe.puro retroviral constructs.

### Biochemical analyses

Cells were stimulated using 0.3 μg/ml αLTβR (a gift from J Browning and A Papandile, Biogen, Cambridge, MA, USA), 100 ng/ml recombinant LTα₁β₂ (Sigma, St. Louis, MO, USA), 1 ng/ml TNF (Roche, BASEL, Switzerland), 1 ng/ml IL-1β (Biosource, Carlsbad, CA, USA), or 1 μg/ml LPS (Enzo, NY, USA), either individually or in combination. EMSA, immunoblot analyses, and IKK assay have been described earlier (*Basak et al., 2007*). Recombinant GST-IκBα (1-54aa) used in IKK assay was from BioBharati Life Sciences, Kolkata, India. NIK was immunoprecipitated (Cell Signaling Technology, Danvers, MA, USA) from cytoplasmic extracts and immunopellets were examined for kinase activity using GST-IκBδ as substrate (GST-p100₄₀₆₋₈₉₉, BioBharati Life Sciences, Kolkata, India). The gel images were acquired using PhosphorImager (GE, Amersham, UK) and quantified in ImageQuant. Immunoblotting of immunoprecipitates was done using TrueBlot (eBioscience, San Diego, CA, USA).

### Gene expression studies

Total RNA was isolated using RNeasy Kit (Qiagen, Venlo, Netherlands). For microarray analysis, labeling, hybridization of RNA samples to the Illumina MouseRef-8 v2.0 Expression BeadChip, data processing and quantile normalization was performed by Sandor Pvt. Ltd (Hyderabad, India). We have

considered genes that are induced at least 1.3 fold by LPS at 24 hr in representative data sets and has a detection p-value < 0.05 for LPS, αLTβR and co-treatment regimes. Next, LPS response genes in WT MEFs were ranked based on the merit of their normalized crosstalk scores, which is defined below and also described earlier (*Zhu et al., 2006*).

Crosstalk score = [(Δco-treatment − (ΔLPS + ΔαLTβR))/0h_int], where 0h_int indicates the signal intensity of a given gene in untreated cells and Δtreatment signifies the differences in signal intensities between treated and untreated cells,

Normalized crosstalk score = Crosstalk Score * [{(ΔL + ΔB))/0h_int/I(ΔL + ΔB))/0h_intl],

As implied, positive crosstalk scores signify hyperactivation, whereas negative crosstalk scores imply diminished gene expressions in the co-treatment regime as compared to cell treatment with the individual stimuli. The ordered gene set was examined in GSEA v2.0.12 (Broad Institute at MIT) (*Subramanian et al., 2005*). The MIAME version of the microarray data set discussed in this publication are available on NCBI Gene Expression Omnibus (accession number GSE62301). For quantitative RT-PCR, total RNA was reverse transcribed with Transcriptor cDNA kit and amplified using Sybr Green PCR mix (Roche, Mannheim, Germany) in ABI7500 cycler. The relative gene expressions were quantified using ΔΔCT method upon normalizing to β-actin mRNA level. Absolute quantification was done using plasmid DNA constructs encoding respective genes as standards and normalized to express as gene/actin mRNA level.

## Murine infection model

Sex matched, 8 to 10 week old mice, fasting for 8 hr, were orally gavaged with $1.2 \times 10^{10}$ cfu of *C. rodentium* strain DBS100 (ATCC 51459). In certain instances, mice were intraperitoneally injected with 200 μg of murine LTβR-IgG1 fusion protein or MOPC21 isotype control (Biogen Idec) 1 day prior to infection, as described (*Wang et al., 2010*). IECs, isolated following published procedure (*Greten et al., 2004*), were utilized for biochemical analyses. For histology, dissected colons were fixed in 10% neutral buffered formalin. Paraffin-embedded tissue sections were stained with anti-myeloperoxidase antibody (Pierce, Waltham, Massachusetts, USA) for neutrophil recruitment or with Hematoxylin and Eosin (H&E) for tissue pathology evaluation. Fecal samples were weighed, homogenized, and serially diluted homogenates were plated on MacConkey agar (HiMedia, Mumbai, India) to score for *C. rodentium*. Similarly, spleens and livers were aseptically removed and assessed for bacterial load. For bone marrow chimera experiment, recipient WT or *Nfkb2*[−/−] mice were lethally irradiated and marrow cells from the indicated donor mice were transferred. After 6–8 weeks, mice were infected.

## Computational modeling

The NF-κB Systems Model v1.0 was simulated in Matlab (v. 2012b, Mathworks, Natick, MA, USA) using the ode15 s (*Basak et al., 2007*). A detailed description of the model has been provided in the Appendix-1. To estimate crosstalk sensitivity, each parameter values were individually increased and decreased by 10%, euclidean distances were used to determine the resultant changes in the crosstalk indexes as compared to the unperturbed system, averaged for a given parameter and normalized to nominal crosstalk index.

## Statistical analysis

Data are expressed as mean of 3–5 quantified biological replicates ± SEM. Statistical significance was calculated by two-tailed Student's *t*-test. For survival curves, log rank (Mantel–Cox) test was conducted.

## Additional files

This article additionally contains (i) 10 figure supplements associated with the main text, (ii) three Supplementary tables (*Supplementary files 1–3*) and five Appendix figures associated with the an Appendix file (Appendix-1), which provides a detailed description of the mathematical model, as well as a file describing Matlab source codes.

## Acknowledgements

We thank Alexander Hoffmann, UCLA for insightful discussions, Gouri Ghosh, UCSD and Satyajit Rath, NII for critical reading of the manuscript. We thank P Nagarajan, SAF and V Kumar, SIL for technical help. This study was supported by an intermediate fellowship to SB from the Wellcome Trust DBT India Alliance and funding from NII-Core. BB and PR thank CSIR for research fellowships.

# Additional information

## Funding

| Funder | Grant reference | Author |
|---|---|---|
| Wellcome Trust | DBT India Alliance, intermediate fellowship | Soumen Basak |
| National Institute of Immunology | Core Funding | Soumen Basak |
| Council of Scientific and Industrial Research (CSIR) | Graduate Student Fellowship | Balaji Banoth, Payel Roy |

The funders had no role in study design, data collection and interpretation, or the decision to submit the work for publication.

## Author contributions

BB, Acquisition of data, Analysis and interpretation of data, Drafting or revising the article; BC, BV, MVRP, PR, Acquisition of data, Analysis and interpretation of data; SB, Conception and design, Analysis and interpretation of data, Drafting or revising the article

## Ethics

Animal experimentation: Wild-type or gene-deficient C57BL/6 mice were housed at NII small animal facility and used strictly in accordance with the Institutional Animal Ethics Committee guidelines of the institute. The protocol was approved by the committee with the approved protocol no: IAEC#258/11 (for embryonic fibroblast cell collection) and IAEC#313/13 (for infection related studies).

# Additional files

## Supplementary files

• Supplementary file 1. List of biochemical species and their initial concentrations in the model.

• Supplementary file 2. List of biochemical reactions and rate constants in the model.

• Supplementary file 3. List of newly described biochemical parameters in the Systems Model v1.0.

## Major dataset

The following dataset was generated:

| Author(s) | Year | Dataset title | Dataset ID and/or URL | Database, license, and accessibility information |
|---|---|---|---|---|
| Banoth B, Basak S | 2014 | Developmental LTbR synergistically activates TLR4 mediated inflammatory RelA/NF-kB response | http://www.ncbi.nlm.nih.gov/geo/query/acc.cgi?token=oncvqccgtlkffap&acc=GSE62301 | Publicly available at the NCBI Gene Expression Omnibus (GSE62301). |

Standard used to collect data: The MIAME version of the microarray data set discussed in this publication are available on NCBI Gene Expression Omnibus (accession number GSE62301).

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

**Appendix-1**

**Appendix-1 includes, a description of the mathematical model, supporting references, and associated *Appendix figures 1–5*, as well as *Supplementary files 1–3*.**

### (i) Description of the mathematical model

The NF-κB Systems Model Version 1.0 builds upon a previously constructed computational model (***Basak et al., 2007***). In the preceding version, canonical IKK2 signals degraded classical IκBs, while non-canonical NIK-IKK1 targeted IκBδ to liberate the RelA:p50 NF-κB dimers, represented as a singular species, into the nucleus. Although, biochemical analyses defined IκBδ as an oligomer of p100, the previous versions represented IκBδ as a composite species that is produced directly from an NF-κB responsive gene. In the current version, we have elaborated biochemical reactions associated with p100 in the model. We have explicitly described p100 as a species that is generated from the transcripts encoded by NF-κB responsive *Nfkb2*. Monomer p100 was subjected to (i) constitutive degradation or (ii) oligomerization to form IκBδ or (iii) NIK-IKK1-dependent processing that generated RelA:p52 dimer as a composite species. In contrast, RelA:p50 dimer was constitutively produced as a singular species. As described in the preceding single dimer model, the activation dynamics of these two RelA NF-κB dimers was coordinately regulated by signal responsive degradation and resynthesis of IκBα, IκBβ, IκBε, and IκBδ during inflammatory signaling. The newly built model consists of 34 species, 138 reactions with 105 parameters; those describe the synthesis and degradation of NF-κB and IκBs, association and dissociation of IκB:NF-κB complexes, nuclear import and export of NF-κB and IκB species as well as processing of p100 (see a detailed wiring in ***Appendix figure 1***). Constrains generated based on our quantitative biochemical analyses (***Appendix figures 2–4***) was used along with published literature and fitting procedures for parameterizing the model (see the description of the parameters in ***Supplementary files 1, 2***). The following assumptions were implicit in the wiring.

1. RelA forms two distinct dimers RelA:p50 and RelA:p52;
2. RelA:p50 dimers are regulated at the level of nuclear translocation;
3. NIK-IKK1 induced processing of p100 generates p52 that forms RelA:p52, which is then regulated by IκBs akin to RelA:p50 dimer.

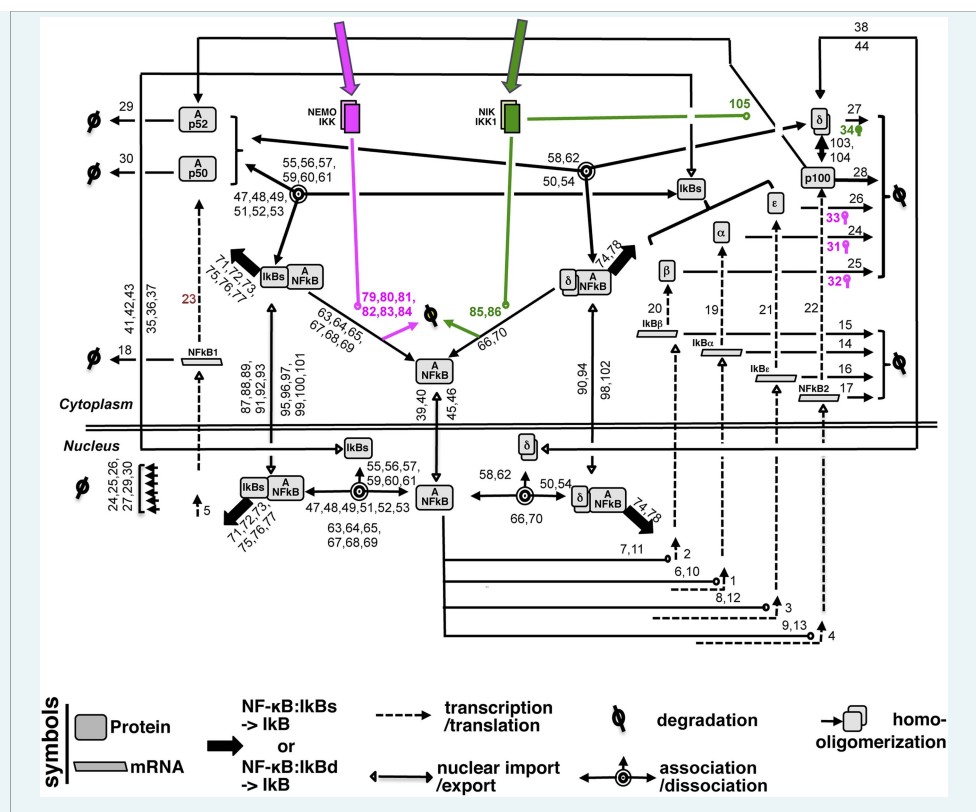

**Appendix figure 1**. A detailed wiring diagram. A wiring diagram depicting the biochemical connectivities between the molecular species included in the NF-κB Systems Model v1.0 to capture signal responsive activation of RelA:p50 (Ap50) and RelA:p52 (Ap52) dimers. The number adjacent to each reaction arrow is the corresponding parameter number, as described in the associated **Supplementary files 1, 2**. For the ease of presentation, three classical IκBs, IκBα, -β and -ε, were presented as IκBs on certain occasions. Similarly, RelA:p50 and RelA:p52 dimers were collectively depicted as A:NFκB.

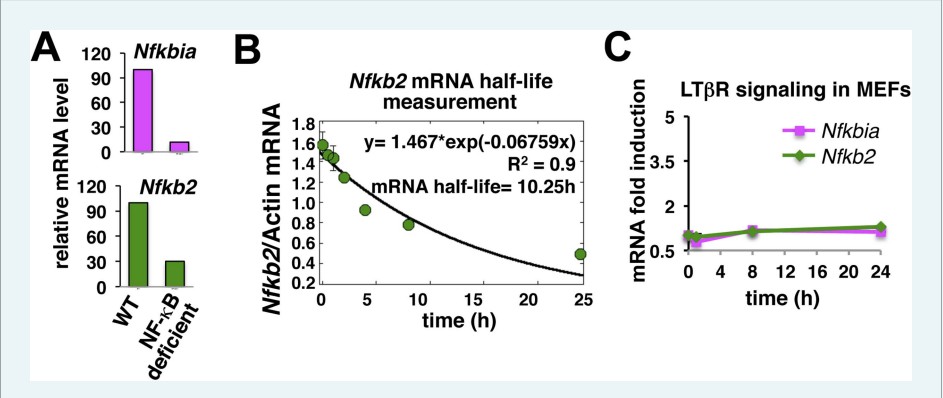

**Appendix figure 2**. *Nfkb2* mRNA analyses in unstimulated cells and during signaling. (**A**) The NF-kB independent constitutive transcription of *Nfkbia* mRNA, which encodes IκBα, and *Nfkb2* mRNA in the late passage MEFs was measured using quantitative RT-PCR. The mRNA levels of *Nfkbia* and *Nfkb2* were reduced to 10% and 25% in NF-κB deficient (*Rela⁻/⁻Rel⁻/⁻Relb⁻/⁻*) cells as compared to WT cells. Transcript level of *b-actin* was used for normalization. (**B**) Quantitative RT-PCR to measure half-life of *Nfkb2* mRNA. WT cells were treated with 5 mg/ml of transcription inhibitor actinomycin-D in a time-course, harvested, absolute mRNA levels of *Nfkb2* and *b-actin* were determined and presented as *Nfkb2* mRNA copy numbers per thousand copies of *b-actin* mRNA. The curve fitting was done assuming first-order decay of *Nfkb2* mRNA to arrive onto a half-life of ~10.5 hr. (**C**) LTβR induced expression of *Nfkbia* and *Nfkb2* mRNAs was

analyzed in a time course. Due to weak NF-κB activation, LTβR activation using 0.3 mg/ml agonist antibody only subtly altered the levels of these two NF-κB target genes in MEFs.

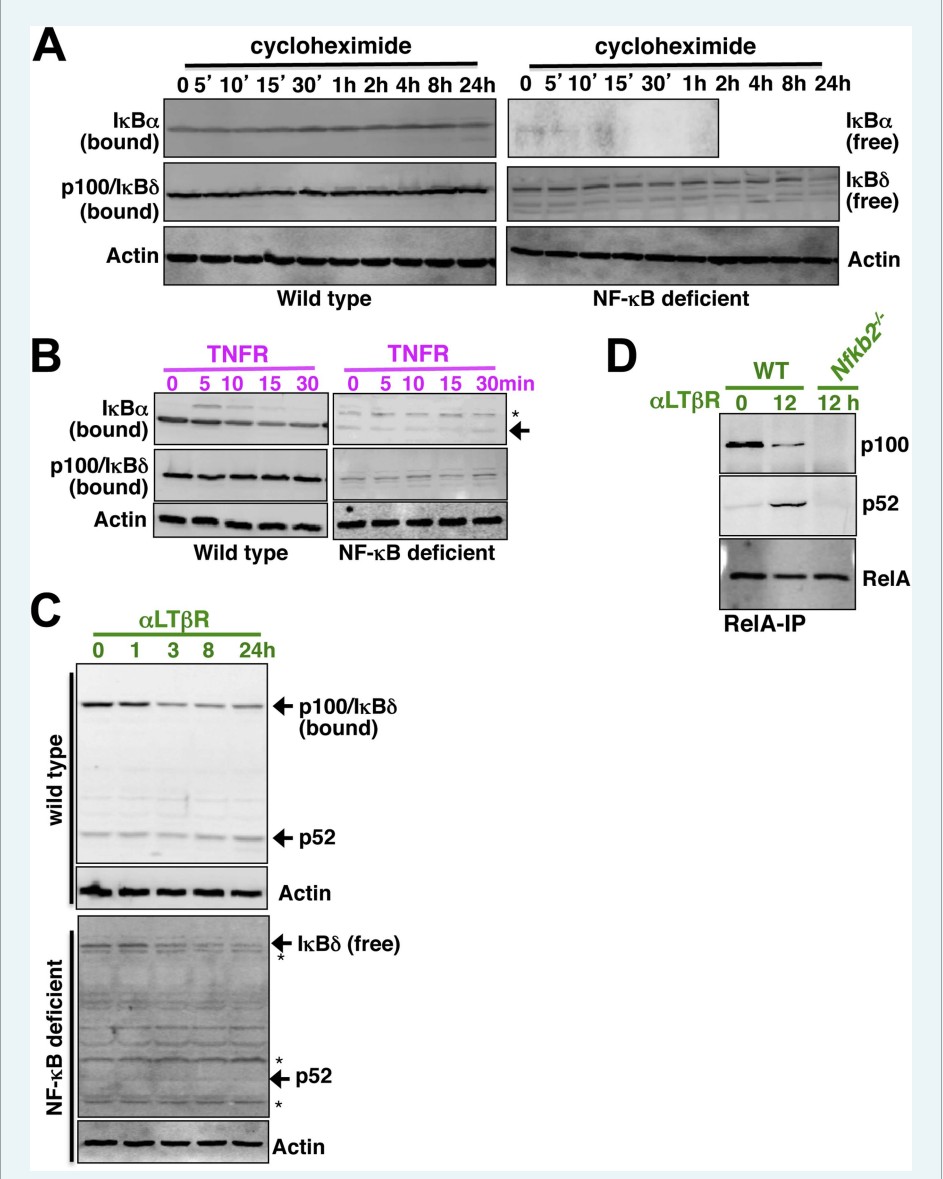

**Appendix figure 3**. Analyzing p52 and p100 protein encoded by *Nfkb2* in unstimulated cells and during signaling. (**A**) Constitutive degradation of NF-κB bound (left panel) or free (right) IκBα (top panel) and p100/IκBδ (middle) were evaluated by immunoblotting respective extracts derived from WT (left) or NF-κB deficient (right) cells treated with 10 mg/ml of protein synthesis inhibitor cycloheximide. As compared to the stable NF-κB bound form, free IκBα degraded rapidly with a half-life ~5 min. In contrast, both NF-κB bound p100/IκBδ present in WT cells (top) and free p100/IκBδ in NF-κB-deficient cells (bottom) were stable with a half-life > 12 hr. Actin (bottom panel) was used as loading control. The data represents two biological replicates. (**B**) Immunoblot revealing TNFR-induced IKK2 mediated degradation of NF-κB bound IκBα in WT (left panel, top) or free IκBα in NF-κB-deficient cells (right, top). Based on these data, IKK2-mediated degradation rates of the free and NF-κB bound classical IκBs were assigned comparable values in the model. Both in its bound (left, middle) or free (right, middle) form, p100/IκBδ were insensitive to IKK2 signals. (**C**) Immunoblot showing LTβR-induced NIK-IKK1 mediated degradation of NF-κB bound (top panel) and free (bottom) p100/IκBδ in WT and NF-κB deficient cells, respectively. Importantly, signal induced p52 accumulation was evident at 24 hr in WT, but not in NF-κB deficient, cells suggesting that

newly generated p52 requires other NF-κB monomers for mutual stabilization. (**D**) Immunoblot of RelA co-immunoprecipitates revealing liberation of RelA from inhibitory p100/IκBδ (top panel) and generation of RelA:p52 complexes (middle) during LTβR signaling. *Nfkb2*−/− cell extracts were used as control (bottom).

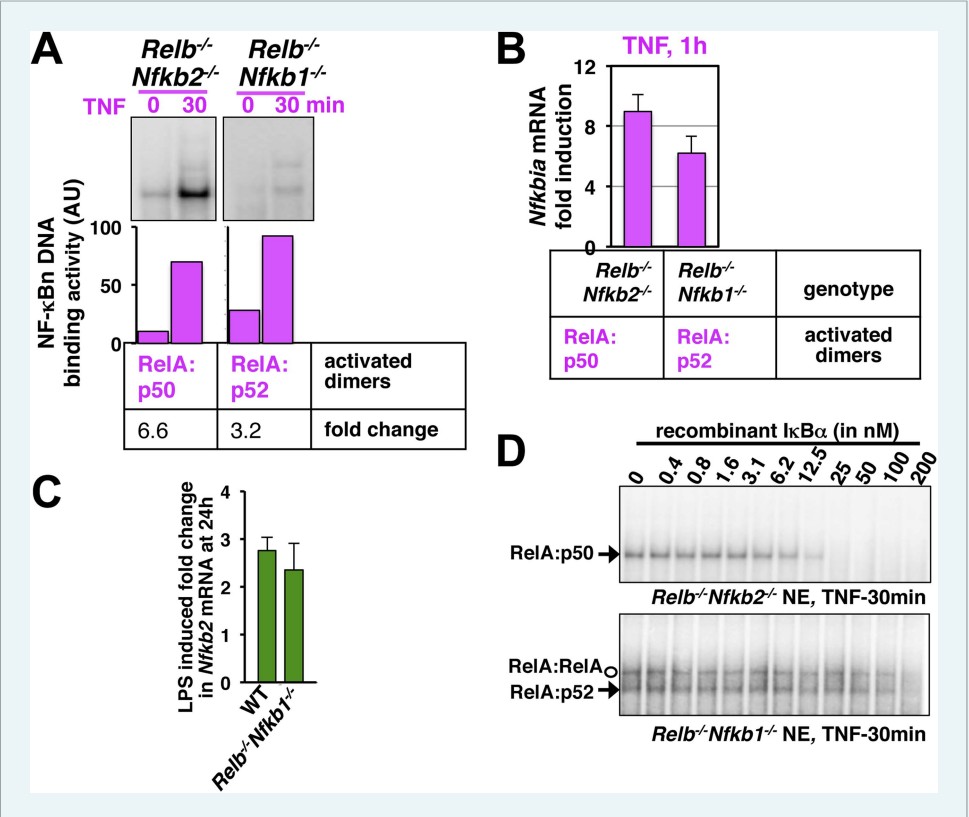

**Appendix figure 4**. Comparing RelA:p50 and RelA:p52 NF-κB dimers. (**A**) EMSA revealing TNF activation of RelA:p50 (left panel) or RelA:p52 (right) dimers in *Relb*−/−*Nfkb2*−/− or *Relb*−/−*Nfkb1*−/− cells, respectively. Quantification of the band intensities (bottom panel) revealed that basal and TNF induced RelA:p52 activities in *Relb*−/−*Nfkb1*−/− are ~10% of the corresponding RelA:p50 activities in *Relb*−/−*Nfkb2*−/−. These values were used as constrains to describe constitutive RelA:p52 dimer generation in WT cells in the model. Both RelA:p50 and RelA:p52 dimers were induced within 30 min of TNF treatment indicating that NEMO-IKK mediated degradation rates of IκBs bound these RelA heterodimers are comparable. Of note, TNF induced fold changes in the activity of the respective RelA dimers were comparable in *Relb*−/−*Nfkb2*−/− and *Relb*−/−*Nfkb1*−/− cells. The result represents three biological replicates. (**B**) NF-κB responsive expression of *Nfkbia* mRNA, encoding IκBα, induced by either RelA:p50 or RelA:p52 dimers in *Relb*−/−*Nfkb2*−/− or *Relb*−/−*Nfkb1*−/− cells upon TNF treatment, respectively. Comparable mRNA fold induction along with our EMSA data suggested similar DNA binding and transcriptional proficiency of RelA:p50 and RelA:p52. Data are expressed as mean of 3 quantified biological replicates ± SEM. (**C**) Gene-expression analyses revealing comparable induction of *Nfkb2* mRNA upon LPS treatment in WT MEFs, those mostly activate RelA:p50 dimer and *Relb*−/−*Nfkb1*−/− MEFs, which exclusively activates RelA:p52 dimer. Our results confirm that RelA:p52 dimer is capable of inducing the expressions of *Nfkb2* mRNA as effectively as RelA:p50 dimers. (**D**) Biochemical analyses indicating weaker IκBα binding to the RelA:p52 dimer. Briefly, nuclear extracts derived from *Relb*−/−*Nfkb2*−/− or *Relb*−/−*Nfkb1*−/− cells treated with TNF was used as a source of free RelA:p50 or RelA:p52, respectively, and normalized to obtain comparable DNA binding activities. Nuclear RelA dimers were incubated with a gradient of recombinant IκBα for 30 min to facilitate the formation of IκB:NF-κB complex, which does not bind DNA in EMSA. The abundance of unbound NF-κB dimers was then scored using EMSA. Incubation with 25 nM IκBα was sufficient to completely inhibit RelA:p50 DNA binding. While, more than 200 nM of IκBα was required to inhibit RelA:p52 and RelA:RelA DNA binding, owing to weaker IκBα binding by these dimers. The data represents three experimental replicates.

Based on the existing literature and our own analyses (*Appendix figures 2–4*), we have assumed that RelA:p50 and RelA:p52 dimers exhibit mostly similar biochemical properties, except for IκBα binding (note *Appendix figure 4D*). By generating RelA:p52 dimer from p100 upon processing, we also captured the mutual stabilization of NF-κB monomers, where, de novo synthesized p52 rescued RelA monomer from degradation to generate RelA:p52 dimer in response to NIK-IKK1 signal. We have summarized the new reactions incorporated into the model version published in *Basak et al. (2007)* to arrive onto the Systems Model v1.0 and provided justification for the parameter values in *Supplementary file 3*.

Simulations were performed in MatLab version 2012b (Mathworks) using the built-in ode15 s solver at default settings. We have introduced a time delay of 65 and 37 min in the NF-κB-induced transcription of *Nfkb2* and those encoding IκBε, respectively, to recapitulate experimental observations. The transcriptional delay function has been executed in the Systems Model v1.0 as follows: for an explicit delay of t1 min, a given NF-κB dependent transcription reaction assumes a NF-κB concentration at basal level up to t1 min. Subsequently, nuclear concentration of NF-κB has been used as such from (t1 + Δt1) min and onwards for computing transcription rates. As previously described, an equilibrium phase was introduced prior to the onset of stimulation phase (*Basak et al., 2007*). Matlab model codes have been provided as an additional source code file with this manuscript. As such, the NF-κB Systems Model version 1.0 was able to reproduce temporal profiles of nuclear RelA:p50 and RelA:p52 dimer activities (*Appendix figure 5*), as measured experimentally in response to TNFR as well as LTβR stimulation (*Figure 1—figure supplement 1*).

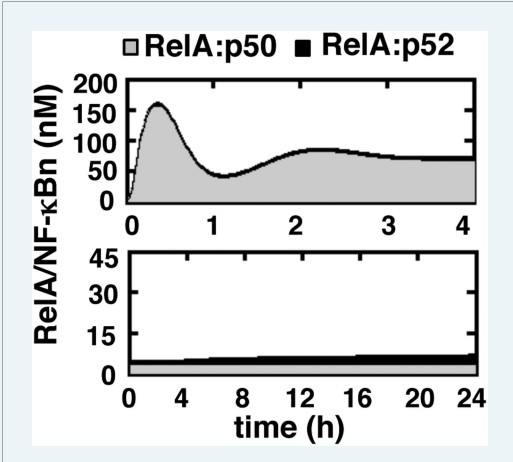

**Appendix figure 5**. Simulating RelA:p50 and RelA:p52 activation in response to canonical and non-canonical signals. Computational simulations of nuclear RelA:p50 (grey) and RelA:p52 (black) induction by TNFR-induced IKK2 (top) or LTβR-induced NIK-IKK1 signals (bottom). Our computational data effectively recapitulate relative abundance of these two RelA dimers during canonical and non-canonical signaling, as experimentally observed and demonstrated in *Figure 1—figure supplement 1*.

