## [Decision Letter]

Thank you for sending your work entitled “Stimulus-selective crosstalk *via* the NF-κB signaling system reinforces innate immune response to alleviate gut infection” for consideration at *eLife*. Your article has been favorably evaluated by Detlef Weigel (Senior editor), a Reviewing editor, and three reviewers.

The Reviewing editor and the reviewers discussed their comments before we reached this decision, and the Reviewing editor has assembled the following comments to help you prepare a revised submission.

Summary:

In this manuscript, the authors describe a potentially novel synergistic effect of LTβR signaling on signaling from LPS-TLR4 that contributes to enhanced NF-kB activity at later time points after stimulation (∼24 hours). As LPS signaling to NF-kB proceeds through the canonical pathway and leads to p50:RelA and p50:c-Rel complexes, while LTβR signaling proceeds through NIK-IKK1 to promote processing of p100 to p52, which then associates with RelB (or is preassociated with it) to form activating p52:RelB dimers, the demonstration that LTβR-induced p52:RelA enhances the canonical pathway, is interesting. As the activation phase of the canonical NF-kB activity is determined by negative feedback through newly synthesized IκBα and IκBδ (as well as molecules such as A20), the ability of LTβR signaling to somehow overcome the observed reduction of canonical NF-kB activity, is also an interesting observation. Some of the data presented are indeed exciting, as is the progression of analysis from computational models to physiology.

The shortcomings of this paper pertain to important deficiencies in quantitation and statistical analysis of results, the lack of some key experimental controls, and there are some questions about the model. The following specific points need to be addressed in order to be considered further for publication.

Specific points:

1) Could the authors base their model more closely on an established model, for example their own [3], or the recent Yilmaz et al. 2014 and then identify very explicitly the reaction and parameters that are new (in separate tables) and justify those values within experimentally determined ranges?

2) Panel 1E is unclear. It seems to indicate that both early and late peaks of NEMO-IKK2 engage in crosstalk, contrary to the claim. This needs clarification.

3) Whereas 1B distinguishes between RelA:p50 and RelA:p52 the later figures do not seem to distinguish them, though this seems to be an important distinction later. The model should be used to make that distinction.

4) Panels within this figure (and others) should be arranged in order or presentation.

5) In Figure 2, experimental data is provided to test the computational predictions. Quantitated line or bar graphs should be shown. As effects are not obvious, statistical significance should be established.

Also, it is not clear why authors use the *Ifnr1*^*-/-*^ MEFs. They previously stated that prolonged NFκB is dues to autocrine TNF. If they are interested in testing this they should use TNF or TNFR, and then the interferon knockouts may be a useful contrast.

Is it possible to show quantitation of late RelA induction in the presence of both signals from biological replicates? This should be included in Figure 2.

The error bars between 3 biological replicates look surprisingly small in Figure 2—figure supplement 1. Why is this?

6) Figure 3 shows transcriptomic analysis, but the analysis is not transparent. The presentation focuses on a crosstalk score, and says that NF-κB target genes are over-represented. How about the genes that are not amplified by the crosstalk? The authors only show LPS here, and it would be more complete in line with the biochemistry if the authors also showed IL1. They then show that this amplification is maintained in *Relb*^*-/-*^ but it is not clear that these are the same genes. Also the negative result with *Relb*^*-/-*^ MEFs demands a positive control where there is a change, such as *Nfkb2*^*-/-*^*.* Later the authors introduce the *Nfkb2*^*-/-*^ MEFs, so maybe the *Relb*^*-/-*^ data could be moved to that figure?

7) In Figure 4, the authors present a one-dimensional sensitivity analysis, but the reliability of this prediction depends on the initial parameterization. Too little information is provided on how the parameterization was achieved and how the error/confidence intervals of experimental data translates to alternative parameter sets, which in turn may affect the results of the sensitivity analysis. Here, given the known regulation by p100 described in the Introduction the authors need no further justification for examining p100 in subsequent panels. Panel B should be quantitated and the three conditions should be run on the same gel, and statistically evaluated. The quantitative analysis could be restricted to the late time point(s). Panels C and D are important, and a quantitation of p50 vs p52 association with RelA should be provided. In B, could the authors add a p50 immunoblot? In C, using both p50 and p52 antibodies should result in ablation of the shift but this control is missing. Basically, we would expect an experimental counterpart for the computational panel F. Panel G is quantitated but a statistical confidence evaluation should be provided. Panel H directly relates to Figure 3 and presumably these experiments were done in parallel. Have the authors also tried to use microarrays and undertaken the analysis shown in Figure 3?

8) Figure 5 is not easy to understand. It would be helpful if the following issues were addressed:

A) Why does panel B show the response to TNF, but everywhere else LPS and IL1 are compared?

B) Why is the *Nfkb2* mRNA fold induction different in panels A and B?

It is not clear what delay-null cells are. If they are simply the transgenic *Nfkb2* described in B, it would be more straightforward to use the same terminology. Panels E/F contain critically important information. They need to examine three conditions: LPS alone, LPS and LTβR, and LTβR alone. Also, there are really 4 cell types that are of interest: WT, *Nfkb2*^*-/-*^*.* and the two transgenes. Distinction should be drawn between RelAp50 and RelAp52. Panel G mentions a positive feedback loop, but the figure does not show any data related to p52. And strictly speaking no data was presented that RelAp52 produces more RelAp52. Finally, synergistic effects shown in Figure 5 occur at very early time points, and are therefore difficult to relate to duration synergy that happens at >8h.

The top diagram seems to be one stimulation condition resulting in the green curve, and the bottom diagram related to the pink curve. Yet the top diagram also contains pink arrows. What makes things difficult to follow is that the canonical and non-canonical arrows come in from different directions, and it is unclear where the curved arrows are pointing.

9) Figure 6 shows exciting data which puts the previous fibroblast studies in physiological context.

The following control might strengthen results (if possible): distinguish between RelAp50 and RelAp52 for panel E, similar to the analysis in 4D, and quantitate the levels of each.

10) Figure 7 suggests that p100 KO's are more susceptible to Citronella infection, but that does not really prove that p52 complexes in LTβR induced synergy are responsible, and instead the p52 complexes might be acting on their own. Also as p100 KO's do not display any general sensitivity to infection, it is unclear whether the Citronella results are unique to these bacteria.

11) The observed effects in Figure 6 and Figure 7 could be 'simply' due to lack of non-canonical NF-κB activation in response to LTβR in the NFκB2^-/-^ strain. The crux of the duration model is that extended RelA activity is RelB-independent. The mouse experiments neither address whether RelA function is involved in pathogen clearance nor do they show that the effects are RelB-independent. How might this question be addressed?

[Editors' note: further revisions were requested prior to acceptance, as described below.]

Thank you for resubmitting your work entitled “Stimulus-selective crosstalk *via* the NF-κB signaling system reinforces innate immune response to alleviate gut infection” for further consideration at *eLife*. Your revised article has been favorably evaluated by Detlef Weigel (Senior editor), a Reviewing editor and one reviewer. The manuscript has been improved but there are some remaining issues pertinent to clarity of presentation that need to be addressed before acceptance, as outlined below:

1) By calling the model NF-κB system model v. 1 it is not clear that the model is based on (and justified by) previous models. The authors should pick a different name (or is there a need for a name given that a model is generally cited with the paper reference?). Similarly the Table should be more explicit. Many values are described as “adapted from [REF]”. If the values are the same, that should be stated. If they were modified then a justification should be given. For some values the authors state “assumed to be similar to [another]”. They should state if they are the same, or when different a justification should be provided. The diagram in Figure 8 could be confusing with many different shapes, colors and shadings. Why not adopt the normal conventions for kinetic model diagrams and indicate the species in the diagram itself rather than the legend? It is hard to keep track of the colors, and indeed it seems that the pink/orange shading in the legend became an orange/red shading in the diagram.

2) In Figure 1 the authors are focused on points that are missing in the range of <2hrs. It is not evident from this figure what curves are missing, and so the figure is difficult to understand. They should find another way to make this point. Why not show the actual crosstalk score in a matrix of IKK2 and IKK1 profiles? That would avoid the inherent problems of applying a threshold (crosstalk proficient) which can also lead to misleading conclusions.

3) The quantitation added to Figure 2 seems surprising in some cases: for example in 2B LPS, the 3 and 5 h points are very different in the gel, but very similar in the graph with minimal error bars. There are several other points that do not transparently seem to square with the gel data. Because the graph is the result of 3 gels, it will not match precisely but the error bars should indicate the true standard deviation in the data. Some clarification of this point is required.

4) Figure 2 is still rather obscure. Why not show the genome-wide data in a more direct manner (e.g. heatmap) so that the quality of it can be assessed. Figure 4 requires a wt control. Does RelB play no role, a minor role?

---

## [Author Response]

*1) Could the authors base their model more closely on an established model, for example their own*
[3], *or the recent Yilmaz et al. 2014 and then identify very explicitly the reaction and parameters that are new (in separate tables) and justify those values within experimentally determined ranges?*

The computational model presented in this manuscript builds on a previously published model (Basak et. al., Cell 2007) to add new details to the pathway involving RelA:p52 dimer with a parameterization supported by additional data (presented in Appendix 1, Figures 9, 10 and 11). We would like to point out that we have already provided a detailed description of the model parameters in the [Supplementary-material SD3-data SD4-data]. As evident from these presentations, the majority of the parameter values are derived from the model versions previously published in [3] (Cell. 2007 Jan 26;128(2):369-81) and to some extent, from [25] (Proc Natl Acad Sci U S A. 2009 Jun16;106(24):9619-2) as well as [26] (Nat Immunol. 2012 Dec;13(12):1162-70).

Following the suggestions by the reviewers, we have now explicitly identified the reactions and parameters that are new in a separate table ([Supplementary-material SD5-data]) and provided justification for each of the parameter values. As presented in Table 3, the parameter values describing the biochemical reactions involving newly described RelA:p52 species was assumed to be similar to that of RelA:p50 dimer. In case of a deviation, such as those involving composite species (*NFκB1*) or RelA:p52 association with IkBs, additional experimental evidence has been provided and also indicated in the table. We hope that our new table will adequately capture the rigor in our model building efforts. Of note, Yilmaz et al. (2014), in their study mostly focused on interdependent p50 and p52 generation and did not model p52 containing DNA binding complexes.

*2) Panel 1E is unclear. It seems to indicate that both early and late peaks of NEMO-IKK2 engage in crosstalk, contrary to the claim. This needs clarification*.

Consistent to our claim, our analyses indeed indicated that NEMO-IKK2 activities with less than 2h of duration are unlikely to engage into crosstalk. Of note, the library used in our experiment consisted of 46 IKK2 inputs with duration ≤ 2h. Yet, none of these short-duration IKK2 activities engaged into signaling crosstalk as reflected in Figure 1 (top panel). In Figure 1—figure supplement 2, we have further verified our claim by using representative short-duration NEMO-IKK2 inputs. Although, the duration threshold for NEMO-IKK2 was somewhat less elaborate as compared to NIK-IKK1, which exhibited a duration threshold of ∼ 8h (bottom panel, Figure 1). We suspect that this difference may have attributed to the undesired confusion undermining NEMO-IKK2 duration control. Notably, this rather narrow time scale in the NEMO-IKK2 duration threshold was effective in discriminating between IL-1R and TLR4 signaling with respect to crosstalk. In the revised text, we have further restructured the pertinent section to highlight the differences between NEMO-IKK2 and NIK-IKK1 activities in the quantum of the duration threshold (in the subsection headed “A duration code controlling crosstalk between canonical and non-canonical NF-κB signaling”).

3) Whereas 1B distinguishes between RelA:p50 and RelA:p52 the later figures do not seem to distinguish them, though this seems to be an important distinction later. The model should be used to make that distinction.

We thank the reviewers for pointing out the apparent logical gap in the figure presentation that may have generated distraction in Figure 1. Although RelA:p52 dimers accounted for only a minor RelA/NF-κB nuclear activity during canonical or non-canonical signaling, the distinction between RelA:p50 and RelA:p52 dimer was made for merely validating the faithfulness of the model in recapitulating experimental data. In the revised draft, instead we have presented the simulation data for total nuclear RelA/NF-κB activity in Figure 1. We agree that such distinction is largely peripheral for confirming the effectiveness of the mathematical model. As also suggested by the reviewers, such distinction by mathematical model instead provided powerful argument in Figure 4 in revealing the plausible mechanism underlying crosstalk control. As discussed later (point [7]), we have now used the mathematical modeling data revealing enhanced level of nuclear RelA:p52 in the crosstalk settings (Figure 4 in the earlier version) in Figure 4 to suggest the involvement of *Nfkb2*-derived RelA:p52 in crosstalk. Accordingly, we have made alterations in the text in the Results section; those did not adversely impact the major conclusions, but offered more streamlined logical progress in the revised manuscript.

*4) Panels within this figure (and others) should be arranged in order or presentation*.

While apologizing for the unintended confusion, we have now arranged all the panels in order of presentation in the respective figures in the revised manuscript.

*5) In*
Figure 2*, experimental data is provided to test the computational predictions. Quantitated line or bar graphs should be shown. As effects are not obvious, statistical significance should be established*.

Our experimental data demonstrated that late RelA NF-κB response to the co-treatment regime is augmented as compared to solitary LPS treatment. Although, the effect on amplitude may appear less dramatic, LTβR cotreatment led to chronically elevated RelA NF-κB activity between 8 -24h as compared to diminishing RelA NF-κB response in solitary TLR4 regime. We argue that the subtitle for the pertinent subsection “Stimulus specific crosstalk allows LTβR signal to prolong TLR4 induced RelA NF-κB response” aptly reflects this effect of LTβ R cotreatment in prolonging TLR4 induced RelA NF-κ B response. As recommended by the reviewers, we have now included the quantitated bar graphs along with statistical tests in Figure 2 in the revised manuscript to further strengthen our claim. These bar diagrams were part of the Figure 2—figure supplement 1 in the initial submission, and now they have been included in the main text. We have also removed the panel merely showing a longer exposure of the EMSA gel in Figure 2 in our revised manuscript to accommodate newer panels in the main figure.

*Also, it is not clear why authors use the* Ifnr1^-/-^
*MEFs. They previously stated that prolonged NFκB is dues to autocrine TNF. If they are interested in testing this they should use TNF or TNFR, and then the interferon knockouts may be a useful contrast*.

Although, the contribution of Trif in prolonging IKK2 activity downstream of TLR4 is well established, whether the autocrine TNF is solely responsible for mediating the effects of Trif remains an active area of research. Indeed, recent studies have established a direct role of Trif in engaging NEMO-IKK2 complex for NF-κ B activation (reviewed in Kawai and Akira, Nature Immunol, 2010). In the revised text, we have now restructured the relevant sentence (in the subsection headed “A duration code controlling crosstalk between canonical and non-canonical NF-κB signaling”) to omit the section unnecessarily emphasizing on TNF autocrine. Given the lack of clarity on function of TNF autocrine, we choose to use *Trif*^*-/-*^, and not *Tnfr1*^*-/-*^, MEFs in our biochemical analyses those only transiently induced NEMO-IKK2 activity downstream of TLR4. However, we agree with the reviewers’ perspective that *Ifnr1*^*-/-*^ does not provide for a powerful contrast with *Trif*^*-/-*^. As the reviewers have also indicated, we concur that *Ifnr1*^*-/-*^ data appears to be superfluous for establishing the duration code and actually could be distracting. Therefore, we have removed the relevant figure panel (Figure 2 in the earlier version) and restructured the text in the revised manuscript (in the subsection headed “Stimulus specific crosstalk allows LTβR signal to prolong TLR4 induced RelA NF-κB response”).

*Is it possible to show quantitation of late RelA induction in the presence of both signals from biological replicates? This should be included in*
Figure 2*.*

As discussed earlier, we have included the quantitated bar graphs along with statistical tests in Figure 2 in the revised manuscript.

*The error bars between 3 biological replicates look surprisingly small in*
Figure 2—figure supplement 1*. Why is this?*

Here, we have examined RelA NF-κB response in the costimulation regime in relation to those induced by solitary TLR4 or LTβ R signals. Given the complex biological question, we have normalized the EMSA signals against the respective IL-1 induced (0.5h) or LPS induced (1h) peak values. As one could anticipate, differences in the normalized data intensities were less obvious, in contrast to the variations in the raw signal intensities normally observed (and we also noted) in different experiments. We have now placed the data quantification (previously presented in Figure 2—figure supplement 1) in Figure 2 and Figure 2 in main text, respectively, in the revised manuscript. In the respective figure legends, we have clearly mentioned the data normalization modalities.

*6)*
Figure 3
*shows transcriptomic analysis, but the analysis is not transparent. The presentation focuses on a crosstalk score, and says that NF-κB target genes are over-represented. How about the genes that are not amplified by the crosstalk? The authors only show LPS here, and it would be more complete in line with the biochemistry if the authors also showed IL1. They then show that this amplification is maintained in* Relb^-/-^
*but it is not clear that these are the same genes. Also the negative result with* Relb^-/-^
*MEFs demands a positive control where there is a change, such as* Nfkb2^-/-^. *Later the authors introduce the* Nfkb2^-/-^*MEFs, so maybe the* Relb^-/-^
*data could be moved to that figure?*

We deeply regret that our transcriptomic analysis did not appear adequately transparent to the reviewers, perhaps owing to the lapses in appropriately communicating the data. We would like to point out that the gene set enrichment analysis (GSEA) is a correlation study, which offered global confirmation for a role of NF-κB in the heightened expressions of LPS induced genes in the costimulation regime. We have provided a detailed description of the gene-expression analyses those include methodologies used for estimating crosstalk scores (included in the Materials and methods in the main text in the revised draft), crosstalk scores of the individual genes ([Supplementary-material SD1-data] in the revised draft) as well as identity of the genes belonging to the NF- κB target gene set used for GSEA ([Supplementary-material SD2-data] in the revised draft). As indicated by the reviewers, we have also noted that several genes were actually downregulated in the costimulation regime. But specific enrichment of NF-κB targets among the hyper-induced genes in GSEA conversely validated that these downregulated genes were less likely to be NF-κB targets. TLR4 signaling, however, activates multiple other transcription factors those include members of ATF, IRF as well as STAT family. At this point, it remains unclear to us if the observed downregulation of gene-expressions is attributed by any of these transcription factors. Owing to a lack of bona fide target gene sets validated through experimental analyses, we were unable to further test the possible engagement of other transcription factors in GSEA. Instead restricting the scope of our conclusion, we could narrowly infer that the augmented RelA NF-κB activity observed in the nucleus in the costimulation regime leads to heightened NF-κB dependent gene-expressions. We have also performed quantitative RT-PCR analyses confirming heightened expressions of several NF-κB target chemokine/cytokine genes in the costimulation regime.

Concurring to the suggestions by the reviewers, we have now performed additional quantitative RT-PCR analyses revealing that LTβR costimulation did not alter IL-1 induced expressions of known NF-κB target chemokine or cytokine genes. This additional data now has been incorporated in Figure 3 in the revised manuscript (in the subsection headed “Signal integration via the NF-κB system amplifies the late expressions of TLR4 induced pro-inflammatory genes”).

Agreeing to the reviewers’ suggestions, we have placed the quantitative RT-PCR data using *Relb*^*-/-*^ MEFs in Figure 4 in the revised manuscript to contrast with the *Nfkb2*^*-/-*^data (in the subsection headed “Non-canonical signal transducer *Nfkb2* supplements RelA:p52 dimer to sustain canonical RelA NF-κB responses”). Given our quantitative RT-PCR analyses revealed an abrogated crosstalk control for several pro-inflammatory, NF-κB target genes, we did not pursue global-scale gene analyses using *Nfkb2*^*-/-*^ MEFs. Of note, GSEA for NF-κB targets is expected to be largely inconsequential in the absence of heightened expressions of genes in the costimulation regime. As indicated later, to maintain the consistency in the data presentation, we have also excluded the microarray data obtained using *Relb*^*-/-*^ MEFs in the revised manuscript and accordingly limited the scope of our conclusion in the revised text.

Despite these changes, the major conclusion that “LTβR costimulation led to the hyperactivation of LPS-induced known RelA target chemokine/cytokine genes in a RelB independent, but *Nfkb2* dependent manner” was preserved. Instead, the revised Figure 4 offered a powerful contrast between the requirement for *Relb* and *Nfkb2* in NF-κB driven gene-expressions during crosstalk.

*7) In*
Figure 4*, the authors present a one-dimensional sensitivity analysis, but the reliability of this prediction depends on the initial parameterization. Too little information is provided on how the parameterization was achieved and how the error/confidence intervals of experimental data translates to alternative parameter sets, which in turn may affect the results of the sensitivity analysis. Here, given the known regulation by p100 described in the Introduction the authors need no further justification for examining p100 in subsequent panels*.

In response to point [1], we have already provided a detailed description of the model parameters in the [Supplementary-material SD3-data SD4-data]. As evident from the table, the majority of the parameter values are derived from the model versions previously published in [3] (Cell. 2007 Jan 26;128(2):369-81) and to some extent, from [25] (Proc Natl Acad Sci U S A. 2009 Jun16;106(24):9619-2) as well as [26] (Nat Immunol. 2012 Dec;13(12):1162- 70). More so, we have now explicitly identified the reaction and parameters that are new in a separate table ([Supplementary-material SD5-data]) and provided justification for the parameter values.

Nevertheless, we apologize that we have not effectively clarified the new findings in our data, which, we submit, are far from being expected. Our sensitivity analyses not only suggested that p100 is important, but also indicated that NF-κB induced transcription of *Nfkb2* might be critical for crosstalk control. We have experimentally validated the importance of p100 in crosstalk control in Figure 4, while we have further examined the significance of NF-κB induced transcription of *Nfkb2* in regulating crosstalk in Figure 5. Importantly, a requirement for NF-κB induced transcription of *Nfkb2* also explained the duration code.

Following the reviewers’ suggestion (point [3]), however, we have restructured the figure panel and used the mathematical modeling data revealing the enhanced level of nuclear RelA:p52 in the crosstalk settings (Figure 4 in the earlier version) along with our sensitivity analyses in Figure 4, respectively. Following this alteration, we could now emphasize in the revised draft that our modeling studies not only predicted an important role of NF-κB induced transcription of *Nfkb2*, but also suggested that *Nfkb2* derived RelA:p52 contributes in augmenting RelA/NF-κB activity during crosstalk (in the subsection headed “Non-canonical signal transducer *Nfkb2* supplements RelA:p52 dimer to sustain canonical RelA NF-κB responses”).

Panel B should be quantitated and the three conditions should be run on the same gel, and statistically evaluated. The quantitative analysis could be restricted to the late time point(s).

Following the reviewers’ suggestion, we have repeated the experiment, analyzed these conditions (untreated and treated for 24h with either LPS or LPS+αLTβR or αLTβR) together in the same gel, and quantitated the relevant band intensities. A plot revealing data quantitation and statistical analyses has now been incorporated in the revised manuscript in Figure 4, while one of the representative immunoblot with all three conditions in the same gel has been provided for the reviewers’ eyes (Figure 13).

Author response image 1.Comparing p100 and p52 levels in the same gel using immunoblot analysis. The data, representative of three biological replicates, consistently reflects augmented accumulation of p52 protein in the costimulation regime as compared to cell treatment with LPS or αLTβR alone for 24h.**DOI:**
http://dx.doi.org/10.7554/eLife.05648.030

*Panels C and D are important, and a quantitation of p50 vs p52 association with RelA should be provided*.

As suggested by the reviewers, we have included the quantification of p50 vs. p52 association with RelA in LPS or LPS+αLT βR or αLTβR treatment regimes in Figure 4 and Figure 4 in the revised manuscript. Our quantification data, which represents 3 biological replicates, consistently reveal a dominant role of p52 *Nfkb2* pathway in crosstalk.

*In B*, *could the authors add a p50 immunoblot?*

In the revised draft, we have included a p50 immunoblot in Figure 4. Further supporting our hypothesis that p52 *Nfkb2* pathway plays a dominant role in crosstalk, we did not notice any discernible change in the p50 level during crosstalk.

*In C, using both p50 and p52 antibodies should result in ablation of the shift but this control is missing*.

The reviewers have asked for an important control. Indeed, on a few occasions, we had used both the antibodies that led to complete ablation of DNA binding activities. In the revised draft, we have included a representative panel in Figure 4 to demonstrate complete shift-ablation upon using both the antibodies. The corresponding full EMSA gel panel has been provided for the reviewers’ eyes (Figure 14).

Author response image 2.Using both p50 and p52 antibodies along with RelB antibody, complete ablation of RelA NF-κB DNA binding activity was achieved in our supershift analyses. The last three lane was used as a panel within Figure 4.**DOI:**
http://dx.doi.org/10.7554/eLife.05648.031

*Basically, we would expect an experimental counterpart for the computational panel F. Panel G is quantitated but a statistical confidence evaluation should be provided*.

We have already provided a statistical test for the data presented in Figure 4 in Figure 2. Adhering to the reviewers’ suggestions, our revised draft now incorporates several quantitative data analyses as well as statistical tests in Figure 4 those significantly helped in fortifying the claim that p52 *Nfkb2* pathway is important for mediating crosstalk effects.

*Panel H directly relates to*
Figure 3
*and presumably these experiments were done in parallel*. *Have the authors also tried to use microarrays and undertaken the analysis shown in*
Figure 3*?*

Given our EMSA as well as quantitative RT-PCR analyses consistently revealed an abrogated crosstalk control in *Nfkb2*^*-/-*^ MEFs, we did not further pursue global scale analyses of NF-κB target gene-expressions in the crosstalk settings using *Nfkb2*^*-/-*^ MEFs. Of note, GSEA for NF-κB targets would be largely inconsequential in the absence of differential gene-expressions. Following the reviewers’ suggestion (point [6]), however, we have restructured the figure panels to include the quantitative RT- PCR data obtained using *Relb*^*-/-*^MEFs (previously in Figure 3) in Figure 4 in the revised draft to contrast with the *Nfkb2*^*-/-*^MEF data. This new figure arrangement strongly supports our conclusion that the crosstalk-amplification of LPS induced NF-κB target gene-expressions requires p52 *Nfkb2* but circumvents the requirement for RelB NF-κB transcription factor. To maintain the consistency in the data presentation, we have removed the microarray data obtained using *Relb*^*-/-*^ MEFs in the revised manuscript. Accordingly, we have stringently focused our conclusion in revised text to suggest that RelB is not required for the heightened induction of pro-inflammatory cytokines in the crosstalk setting. Our conclusion is in line with the overall proposal that crosstalk potentiates innate immune response to bacterial pathogens by prolonging pro-inflammatory gene-expressions through RelA.

*8)*
Figure 5
*is not easy to understand. It would be helpful if the following issues were addressed*:

A) Why does panel B show the response to TNF, but everywhere else LPS and IL1 are compared?

We sincerely apologise for the lack of clarity in describing Figure 5 and we have implemented several changes in the relevant result section in the text to offer further clarity.

TNFR1 induced NF-κB signaling has been traditionally considered as a bona fide representative of canonical signaling. Indeed, we have used examples of TNFR1 singling in Figure 1 to claim that our model was able to recapitulate NF-κB activation *via* the canonical pathway. Although, we have discovered an important role of the promoter intrinsic delay in *Nfkb2* mRNA induction in crosstalk regulations, the delayed induction per se was not a novel finding. In this context, we have presented the TNF data and referred to the earlier research, which originally articulated the delay function (Basak et al., Cell 2007). In the revised text, we have attempted to further clarify this issue (in the subsection entitled “Inducible synthesis of *Nfkb2* by canonical signal triggers a positive feedback loop during crosstalk”).

*B) Why is the* Nfkb2 *mRNA fold induction different in panels A and B?*

We suspect that individual doses of LPS or TNF used in this study may have resulted in dissimilar induction levels of *Nfkb2* mRNA. Of note, we have routinely used 1µg/ml of LPS and 1ng/ml of TNF those led to unequal levels of nuclear NF-κ B activity. It is also possible that the observed stimulus specific variations are attributed by other mechanisms involving chromatin modifications.

*It is not clear what delay-null cells are. If they are simply the transgenic* Nfkb2 *described in B, it would be more straightforward to use the same terminology*.

Our computational simulations predicted that NF-κB inducible synthesis of *Nfkb2* is important for mediating crosstalk between TLR4 and αLTβR signals. To experimentally validate the prediction, we have engineered *Nfkb2*^*-/-*^ cell-line for transgenic (retroviral) expression of *Nfkb2* from either constitutive or NF-κB responsive promoter (Figure 5). Unlike the observed delay in the inducible synthesis from the endogenous promoter, *Nfkb2* mRNA was readily induced when expressed from the NF-κB responsive exogenous promoter. Indeed, the delay null cells represent the engineered *Nfkb2*^*-/-*^ cells, which inducibly express *Nfkb2* mRNA without any delay. Following reviewers’ suggestion, we have now unambiguously denoted these two cell-lines as *Nfkb2*^*-/-*^ cell-lines expressing *Nfkb2*-Tg either constitutively or inducibly. We have now indicated in the text that inducible transgene expresses *Nfkb2* mRNA without any delay (in the subsection “Inducible synthesis of *Nfkb2* by canonical signal triggers a positive feedback loop during crosstalk”). We thank the reviewers, as these changes are likely to bring more clarity in the text.

*Panels E/F contain critically important information. They need to examine three conditions: LPS alone, LPS and LTβR, and LTβR alone. Also, there are really 4 cell types that are of interest: WT,* Nfkb2^-/-^, *and the two transgenes. Distinction should be drawn between RelAp50 and RelAp52*.

Following the reviewers’ suggestions, we have performed additional experiments to compare solitary LPS treatment, solitary αLTβR treatment and LPS+ αLTβR costimulation conditions. In Figure 5, we have compared two engineered *Nfkb2*^*-/-*^ cell-lines, those express *Nfkb2* transgene from either constitutive or NF-κB responsive promoter, respectively. First, we confirmed a lack of LPS inducible expression of *Nfkb2* from the constitutive promoter, but 3.5 fold induced expression from the inducible promoter (Figure 5—figure supplement 1, additional data). Second, additional studies could convincingly demonstrate that constitutive expression of *Nfkb2* abrogates crosstalk-amplification of TLR4 induced late NF-κB activity by concomitant LTβR signal and that NF-κB inducible expression of *Nfkb2* is required (Figure 5 in the revised draft, additional data). Of note, LTβR response per se was not considerably different in these two cell-lines.

Furthermore, our supershift analyses revealed that inducible expression of *Nfkb2* transgene augments RelA/NF-κB activation in the crosstalk settings by potentiating RelA:p52 dimer induction (Figure 5 in the revised draft, additional data). Indeed, these additional analyses supported our hypothesis that NF-κB inducible synthesis of *Nfkb2* potentiates RelA:p52 induction to generate crosstalk at the level of RelA/NF-κB activation. We have dedicated Figure 4 to rigorously compare WT and *Nfkb2*^*-/-*^ cells.

*Panel G mentions a positive feedback loop, but the figure does not show any data related to p52. And strictly speaking no data was presented that RelAp52 produces more RelAp52*.

In panel G (panel H in the resubmitted draft), we have summarized the insight obtained from our mechanistic studies (Figures 1, 2, 3, 4 and 5) in a schematic presentation. We have revealed an involvement of *Nfkb2* derived p52 in crosstalk in Figure 4. In a suitably sub-titled result section, we have further examined the requirement for inducible synthesis of *Nfkb2* in crosstalk in Figure 5. In response to reviewers’ concern, we have now incorporated additional experimental evidence in the revised manuscript demonstrating a role of RelA:p52 in crosstalk in Figure 5.

Several recent studies have strongly suggested that RelA containing heterodimers share DNA binding and gene-expression specificities (Siggers et al., Nature Immunol 2011; Zhao et al., Cell Reports 2014). We have also shown that RelA:p50 and RelA:p52 dimers were equally efficient in inducing *Nfkbia* mRNA expressions during TNFR1 signaling (Appendix 1, Figure 11). Now in the revised manuscript, we further reveal comparable efficiency of the RelA:p52 dimer in inducing *Nfkb2* mRNA during TLR4 signaling using *Relb*^*-/-*^*Nfkb1*^*-/-*^ MEFs (Appendix 1, Figure 11).

*Finally, synergistic effects shown in*
Figure 5
*occur at very early time points, and are therefore difficult to relate to duration synergy that happens at >8h*.

TLR4 signal induces *Nfkb2* mRNA expressions in WT MEFs, albeit with a delay (Figure 5). Our studies indicated that NIK-IKK1 signal targets the late acting p100 *Nfkb2* feedback for generating crosstalk effect at 8-24h in LPS+αLT βR costimulation regime (Figure 2). In the Discussion section, we have elaborated a requirement for the convergence of canonical and non-canonical signals in mediating crosstalk.

The above-mentioned scenario is truly different from crosstalk observed in the IL-1 regime in the engineered cell-line. Unlike late induction of *Nfkb2* mRNA in WT MEFs by LPS, IL-1 signal led to early induction in the engineered cells, which inducibly express *Nfkb2* transgene without the delay (Figure 5). Pretreatment of these engineered cells with αLTβR for 8h and subsequent IL -1 treatment effectively converged the non-canonical signal to IL-1 induced *Nfkb2* feedback, thereby, potentiating p52 production (Figure 5—figure supplement 1) and prolonging IL-1 induced RelA response (Figure 5 and Figure 5—figure supplement 1). Correlating with the early onset of *Nfkb2* mRNA induction in response to IL-1 treatment, observed crosstalk effects were obvious within 1h of cotreatment in these cells. In the revised manuscript, we have attempted to further clarify this issue (please see the subsection entitled “Inducible synthesis of *Nfkb2* by canonical signal triggers a positive feedback loop during crosstalk”).

*The top diagram seems to be one stimulation condition resulting in the green curve, and the bottom diagram related to the pink curve. Yet the top diagram also contains pink arrows. What makes things difficult to follow is that the canonical and non-canonical arrows come in from different directions, and it is unclear where the curved arrows are pointing*.

We thank the reviewers for pointing out the apparent confusions in the graphical presentation in Figure 5. In the revised manuscript, we have now used gray color to depict biomolecular species and intracellular biochemical reactions those engage during canonical or non-canonical signaling. We have also used magenta color to depict canonical signaling inputs, green color to represent non-canonical inputs. In the revised manuscript including in Figure 5, we have now consistently presented the cellular responses to solitary canonical signals using magenta, those to singular non-canonical inducers in green and cellular activities in response to costimulation regimes were articulated using black. Replacing the curved arrows, we have now used pointed arrows to connect RelA to the respective temporal activity profiles representing NF-κB responses to either solitary canonical signal or costimulation.

*9)*
Figure 6
*shows exciting data which puts the previous fibroblast studies in physiological context*.

*The following control might strengthen results (if possible): distinguish between RelAp50 and RelAp52 for panel E, similar to the analysis in 4D, and quantitate the levels of each*.

We thank the reviewers for their excitement and support. Following the reviewers’ suggestion, we have repeated the supershift analyses using nuclear extracts derived from IECs. Our additional experiments revealed considerable nuclear accumulation of RelA:p52 dimer, along with RelA:p50, in the enterocytes at day5 post-infection (Figure 6—figure supplement 1 in the revised manuscript, see revised text in the subsection headed “*Nfkb2* integrates lymphotoxin signal within intestinal niche to reinforce epithelial NF-κB responses to *Citrobacter rodentium*”). As pointed out later, it also seems that our previous supplementary data showing the absence of RelB containing NF-κB dimers in enterocytes was not adequately noticed. Therefore, we have presented a supershift panel in the main text in Figure 6 revealing that nuclear NF-κB activity is exclusively composed of RelA dimers in IECs.

*10)*
Figure 7
*suggests that p100 KO's are more susceptible to Citronella infection, but that does not really prove that p52 complexes in LTβR induced synergy are responsible, and instead the p52 complexes might be acting on their own*.

Solitary LTβR stimulation generates RelB:p52 dimer, while our mechanistic studies (Figures 1, 2, 3, 4 and 5) suggested that convergence of LTβR and TLR4 signals generates an additional RelA:p52 activity in a synergistic manner. We assume that the reviewers may have enquired about the possible engagement of RelB:p52 complexes, which is generated independent of crosstalk regulations, in pathogen response. Using colon derived MSIE cells, we could show that solitary LTβR stimulation only weakly activates NF-κB signaling (Figure 6). But as a co-stimulus, LTβR signal significantly augmented TLR4 induced RelA activity in these intestinal epithelial cells. Furthermore, our supershift analyses clearly demonstrated a complete absence of RelB DNA binding complexes in the enterocytes derived from infected WT mice. Given the importance of the question, we have repeated the supershift analyses and incorporated the supershift data in the revised main text in Figure 6 revealing that nuclear NF-κB activity is exclusively composed of RelA dimers in IECs. Our additional experiments also revealed considerable nuclear accumulation of RelA:p52 dimer along with RelA:p50 in the enterocytes (Figure 6—figure supplement 1). Taken together, these results suggested that the observed defect in the innate immune response to Citrobacter infection (Figure 6) in *Nfkb2*^*- /-*^ mice unlikely involve epithelial RelB activity. Notably, our analyses also confirmed a critical requirement of *Nfkb2* in the non-hematopoietic cells in imparting resilience to pathogen infection. Although, we agree that the observed mortality (Figure 7) could be, in fact, a consequence of deficiencies in the several other cell compartments and may include crosstalk independent *Nfkb2* functions. As discussed in the Results section (please see the subheading “Stromal expression of *Nfkb2* is required for limiting *C. rodentium* infection”) and in the Discussion, we also do not rule out crosstalk-independent engagement of the *Nfkb2* pathway in other cell-types at a later stage of infection, those may contribute to the observed mortality in *Nfkb2*^-/-^ mice. However, epithelial RelA NF-κB activation defects coupled to aggravated early colon pathology and early onset of mortality in *Nfkb2*^-/-^ mice suggested that the stromal requirement of *Nfkb2*, at least in part, lies within the intestinal epithelial cells in the initial events controlling early innate immunity and involves crosstalk regulations. In the revised manuscript, we have subtly altered the text in the relevant Results section discussing Figure 7 to address the point raised by the reviewers.

*Also as p100 KO's do not display any general sensitivity to infection, it is unclear whether the Citronella results are unique to these bacteria*.

In an accompanying unpublished study, we are addressing the cell-type specificity of signaling crosstalk between LTβR and TLR4. Our analyses established that such crosstalk regulations are less likely to be important in myelomonocytic cells. A related manuscript articulating the mechanism underlying cell-type specificity of crosstalk control is currently in preparation. While these results offer an explanation for the apparent lack of general sensitivity in *Nfkb2*^*-/-*^ mice, additional studies will be important to further examine the immune responses in *Nfkb2*^*-/-*^ mice to other microbial pathogens, particularly those infected through the gut. We have now discussed this issue in the Discussion.

*11) The observed effects in*
Figure 6
*and*
Figure 7
*could be 'simply' due to lack of non-canonical NF-κB activation in response to LTβR in the NFκB2*^*-/-*^
*strain*.

The point [11] raised by the reviewers is not very different from the point [10] raised earlier. Unlike the mechanistic studies in the cell culture system (Figures 1, 2, 3, 4 and 5), directly linking *Nfkb2* dependent RelA activity (Figure 6) with the mouse phenotype (Figure 7) is somewhat daunting. We could ascertain in Figure 6 that crosstalk amplification of epithelial RelA response by *Nfkb2* potentiates the expressions of neutrophil-attracting chemokine and cytokine genes. We have gladly noted that the reviewers have lauded our experimental approach and agreed with the interpretation of the data [point 9]. In Figure 6 (Figure 6 in the revised draft), we have simply concluded that defective epithelial expressions of chemokine and cytokine genes may have led to diminished neutrophil recruitment.

We would also like to also point out that prior analysis with *Nfkb2*^*-/-*^ mice revealed only mild lymph node phenotype (Lo et al., Blood 2006), unlike a complete lack of lymph nodes in *ltbr*^*--/-*^ mice. Subsequent mechanistic studies demonstrated that constitutive RelB:p50 activity largely compensates for the absence of LTβR induced RelB:p52 activity in this knockout. These results suggest that the defect in non- canonical RelB:p52 activation in response to LTβR is tolerated in Nfkb2^-/-^ system. Although, we do not rule out crosstalk-independent engagement of the *Nfkb2* pathway in other cell-types at a later stage of infection, those may contribute to the observed mortality in *Nfkb2*^-/-^ mice. In the revised text, we have attempted to further clarify this point and toned down the interpretation of our mortality data (in the subsection headed “Stromal expression of *Nfkb2* is required for limiting *C. rodentium* infection”).

The crux of the duration model is that extended RelA activity is RelB-independent. The mouse experiments neither address whether RelA function is involved in pathogen clearance nor do they show that the effects are RelB-independent. How might this question be addressed?

In general, a critical role of RelA/NF-κB in the pro-inflammatory gene-expressions is firmly established. Indeed, genetic studies have confirmed a requirement of RelA in the epithelial expressions of chemokine and cytokine genes and consequent neutrophil recruitment in LPS induced pneumonia (Alcamo et al., J. Immunol., 2001). In the revised manuscript, we have now cited this important contribution in the subsection “*Nfkb2* integrates lymphotoxin signal within intestinal niche to reinforce epithelial NF-κB responses to *Citrobacter rodentium*”. Furthermore, TLR4 activated RelA was implicated in the chemokine gene expressions and neutrophil recruitment upon bacterial infections (Khan et al., Infect. Immunity., 2006, discussed in the second paragraph of the Discussion section). Consistent to these reports, our study revealed a correlation between diminished RelA activity in IECs and the inability of *Nfkb2*^*-/-*^ mice in limiting bacterial infection. Furthermore, our supershift analyses clearly demonstrated a complete absence of RelB DNA binding complexes in the enterocytes derived from infected WT mice (Figure 6). These results strongly suggested that epithelial NF-κ B response to pathogen infection is mediated by RelA in a RelB independent manner. Of note, pre-existing conditions in *Relb*
^*-/-*^mice with signatures of multiorgan inflammation and a complete lack of secondary lymph nodes (Weih et al, Cell 1995) deterred us from further analyzing these mice in Citrobacter infection experiments. In light of these constructive criticisms and concerns, however, we have felt the necessity to moderate the interpretation of our *in vivo* data. Accordingly, we have emphasized the correlative nature of our argument in the aforementioned subsection and indicated a possible role of RelB in other cell-types in pathogen response.

[Editors' note: further revisions were requested prior to acceptance, as described below.]

*1) By calling the model NF-κB system model v. 1 it is not clear that the model is based on (and justified by) previous models. The authors should pick a different name (or is there a need for a name given that a model is generally cited with the paper reference?)*.

We are fully aware of elegant model building efforts of other NF-κB research groups. While apologizing, we certainly did not intend to suppress that the core circuitry as well as the parameter space of our current NF-κB multidimer model was largely based upon previously published single dimer model versions (3; 25). However, we would like to point out that the scope and the detailed wiring of our current model is significantly different. As such, the NF-κB Systems model *v.*1 describes two different RelA NF-κB heterodimers, but as composite species. In an iterative model building effort, my laboratory has now implemented another advanced NF-κB model (The NF-κB systems model *v*.2), which depicts dimerization of the NF-κB monomers forming various heterodimers. In the future, we hope to expand the scope of the model to arrive at the NF-κB systems model *v*.3, which would address issues pertinent to homodimeric NF-κB species. We feel that it is somewhat important to name these model versions generated in a research program to maintain a catalogue of these incremental advances. Although a detailed description with relevant references has been presented in Appendix 1, we have now also referred to those preceding model versions at the first occurrence of the NF-κB systems Model *v*.1 to avoid further confusion (in the subsection entitled “A duration code controlling crosstalk between canonical and non-canonical NF-κB signaling”).

*Similarly the Table should be more explicit. Many values are described as* “*adapted from [REF]*”*. If the values are the same, that should be stated. If they were modified then a justification should be given. For some values the authors state* “*assumed to be similar to [another]*”*. They should state if they are the same, or when different a justification should be provided*.

Indeed, the terminology “adapted from” indicates that the parameter values were derived from another model version, but were subjected to minor modifications (fitting) for adapting to the newer wiring in the NF-κB Systems Model version 1.0. Concurring to the suggestion, we have now modified the table presented in [Supplementary-material SD4-data] to indicate the quantum of modification with proper justification. Also in the modified table presented in [Supplementary-material SD5-data], we have now explicitly mentioned if the values were identical. In case of a deviation, we have provided justification using our biochemical data. We have now included a summary note in [Supplementary-material SD4-data] to highlight the changes.

As such, out of the total 105-parameter values ([Supplementary-material SD4-data]), 34 were identical to those published in earlier model versions. Moreover, 20 were derived from the published literature, but were subjected to a minor < 3 fold modification for adapting to the NF-κB Systems Model version 1.0. Another 4 parameters were modified < 5 fold for fitting. For additional 12 parameters, further experimental evidence was provided to justify the alterations. Furthermore, another 27 parameters related to the newly described RelA:p52 dimer ([Supplementary-material SD4-data] and [Supplementary-material SD5-data]) were assumed to be identical to those of the RelA:p50 dimer, and the assumption was justified using our own experimental measurements and literature. An exception was being made for the association rates underlying RelA:p52-IkB complex formation (a total of 8 parameters) basing on our experimental analyses.

*The diagram in*
Figure 8
*could be confusing with many different shapes, colors and shadings. Why not adopt the normal conventions for kinetic model diagrams and indicate the species in the diagram itself rather than the legend? It is hard to keep track of the colors, and indeed it seems that the pink/orange shading in the legend became an orange/red shading in the diagram.*

Accepting the constructive criticism, we have now presented the diagram adhering to the formalism followed in Figure 5. Not only, we have eliminated the color scheme, but also specified the species names in the diagram.

*2) In*
Figure 1
*the authors are focused on points that are missing in the range of <2hrs. It is not evident from this figure what curves are missing, and so the figure is difficult to understand. They should find another way to make this point. Why not show the actual crosstalk score in a matrix of IKK2 and IKK1 profiles? That would avoid the inherent problems of applying a threshold (crosstalk proficient) which can also lead to misleading conclusions*.

For the past few years, we have been discussing the possible mode of presentation of the duration code data with several of our colleagues. As such, for a given duration of IKK2 or IKK1 (a+b), there are multiple kinase activities within the library those differ with respect to peak amplitude (h) and peak onset time (a) (see Figure 1—figure supplement 2). For a specific IKK2 activity explicitly defined through peak duration, amplitude and onset time, again all possible IKK1 activities must be considered for evaluating its crosstalk potency. We argue that hierarchical clustering of kinase profiles in a given axis would be even more arbitrary. Therefore, we contest that it is not possible to present crosstalk score in a matrix of IKK2 and IKK1 profiles in a two-dimensional plot as suggested. Unfortunately, even a three-dimensional plot would be insufficient. To remain connected to a broader audience (the manuscript describes bacterial colitis studies), we did not consider other multidimensional plotting modalities. Instead, we have attempted to reduce the complexity of the data by analyzing top 10% crosstalk proficient IKK2 or IKK1 profiles separately for peak duration or amplitude. In Figure 1—figure supplement 2, we have further utilized candidate kinase activity profiles in a case study to confirm fidelity of our interpretation. Moreover, use of experimentally derived long-duration or short-duration kinase activities consistently confirmed the presence of a duration code (Figure 1). With these arguments, we hope to be able to convince you of our rigor in ensuring that threshold intrinsic issues did not lead to erroneous conclusions in our study.

*3) The quantitation added to*
Figure 2
*seems surprising in some cases: for example in 2B LPS, the 3 and 5 h points are very different in the gel, but very similar in the graph with minimal error bars. There are several other points that do not transparently seem to square with the gel data. Because the graph is the result of 3 gels, it will not match precisely but the error bars should indicate the true standard deviation in the data. Some clarification of this point is required*.

As described in Materials and methods, gel images were acquired using PhosphorImager (GE, Amersham, UK) and quantified in ImageQuant software. We agree that the reduction in NFκB activity at 3h was less obvious in the quantitation panel as compared to the presented gel picture. As suggested by the reviewers, the quantitation panel reflects an average of 3 experiments. Note error bar for the 3h time-point is largest among all LPS time-points that indicate variations in the 3h data point in replicate experiments. As explained earlier in response to reviewers’ comments, we have normalized EMSA signals against respective IL-1 induced (0.5h) or LPS induced (1h) peak values. As one could anticipate, our normalization procedure led to lesser-pronounced error bars, including those for 3h LPS data point. In the revised draft, we have now separately indicated data normalization procedure for LPS regime (see Figure legend to Figure 2). As such, 3h LPS time-point is also peripheral to our principal claim that LTβR costimulation augments LPS induced late NFκB activity.

*4)*
Figure 2
*is still rather obscure. Why not show the genome-wide data in a more direct manner (e.g. heatmap) so that the quality of it can be assessed*.

We would like to point out that our experimental settings compare three different cell-conditions, viz. LPS stimulated, αLTβR treated and costimulated. We would also like to emphasize that the goal of our genome-wide analyses was to identify a set of genes those are synergistically activated, and not merely hyperinduced, in the costimulation regime. Although suitable for presenting differential gene-expressions for a given stimulation regime, traditional heatmaps, with very small dynamic range, are inept in capturing synergistic gene-activations that require simultaneous and complex comparisons across multiple stimulation regimes. Therefore, we have adapted a previously published methodology (see ref. [44]), which enables comparison of multiple stimulation regimes. Indeed, our crosstalk score based analyses identified 114 genes those are not only hyperinduced, but synergistically activated in the costimulation regime as compared to solitary LPS or α LTβR treatments (see bottom panel, Figure 3). More so, heatmap based analyses are not expected to disclose if synergistically activated genes are also likely to be NF-κB targets. Based on the literature survey and our discussions with several experts, we trust that GSEA (see ref. [33]) offers best possible solution to the complex problem, which we have addressed. Note, we have referred to the bottom panel of Figure 3 in the text while concluding synergistic gene-effects (please see “Signal integration via the NF-κB system amplifies the late expressions of TLR4 induced pro-inflammatory genes”). In essence, we submit that Figure 3 is as such more informative, in conjunction with the detailed methodologies, than a mere heatmap presentation.

Figure 4
*requires a wt control. Does RelB play no role, a minor role?*

As you may recall, we initially had both WT and *Relb*^*-/-*^ gene-expression data in Figure 3. Following reviewers’ suggestions, we moved *Relb*^*-/-*^ gene-expression data in Figure 4 in the revised manuscript to offer a contrast with gene-expressions scored in *Nfkb2*^*-/-*^. Careful inspection of our WT (Figure 3) and *Relb*^*-/-*^ gene-expression data (Figure 4) revealed that crosstalk-amplification of LPS induced gene-expressions are largely intact for IL1β, RANTES and IP-10. However, we have noted that crosstalk effects on MIP-1α expressions are rather muted in *Relb*^*- /-*^ MEFs owing to prolonged expressions of this gene in response to solitary LPS treatment in the absence of RelB. While insisting that our Figure 3 provides for a control for Figure 4 as far as the role of RelB is concerned, we have now revised the text to indicate a possible minor (although indirect) role of RelB (please see the end of the subsection headed “Non-canonical signal transducer *Nfkb2* supplements RelA:p52 dimer to sustain canonical RelA NF-κB responses”).